# MGAN: Training Generative Adversarial Nets with Multiple Generators

**Quan Hoang**
University of Massachusetts-Amherst
Amherst, MA, USA
qhoang@umass.edu

**Tu Dinh Nguyen, Trung Le, Dinh Phung**
PRaDA Centre, Deakin University
Geelong, Australia
{tu.nguyen,trung.l,dinh.phung}
@deakin.edu.au

## Abstract

We propose in this paper a new approach to train the Generative Adversarial Nets (GANs) with a mixture of generators to overcome the mode collapsing problem. The main intuition is to employ multiple generators, instead of using a single one as in the original GAN. The idea is simple, yet proven to be extremely effective at covering diverse data modes, easily overcoming the mode collapsing problem and delivering state-of-the-art results. A minimax formulation was able to establish among a classifier, a discriminator, and a set of generators in a similar spirit with GAN. Generators create samples that are intended to come from the same distribution as the training data, whilst the discriminator determines whether samples are true data or generated by generators, and the classifier specifies which generator a sample comes from. The distinguishing feature is that internal samples are created from multiple generators, and then one of them will be randomly selected as final output similar to the mechanism of a probabilistic mixture model. We term our method *Mixture Generative Adversarial Nets* (MGAN). We develop theoretical analysis to prove that, at the equilibrium, the Jensen-Shannon divergence (JSD) between the mixture of generators' distributions and the empirical data distribution is minimal, whilst the JSD among generators' distributions is maximal, hence effectively avoiding the mode collapsing problem. By utilizing parameter sharing, our proposed model adds minimal computational cost to the standard GAN, and thus can also efficiently scale to large-scale datasets. We conduct extensive experiments on synthetic 2D data and natural image databases (CIFAR-10, STL-10 and ImageNet) to demonstrate the superior performance of our MGAN in achieving state-of-the-art Inception scores over latest baselines, generating diverse and appealing recognizable objects at different resolutions, and specializing in capturing different types of objects by the generators.

## 1 Introduction

Generative Adversarial Nets (GANs) (Goodfellow et al., 2014) are a recent novel class of deep generative models that are successfully applied to a large variety of applications such as image, video generation, image inpainting, semantic segmentation, image-to-image translation, and text-to-image synthesis, to name a few (Goodfellow, 2016). From the game theory metaphor, the model consists of a discriminator and a generator playing a two-player minimax game, wherein the generator aims to generate samples that resemble those in the training data whilst the discriminator tries to distinguish between the two as narrated in (Goodfellow et al., 2014). Training GAN, however, is challenging as it can be easily trapped into the mode collapsing problem where the generator only concentrates on producing samples lying on a few modes instead of the whole data space (Goodfellow, 2016).

Many GAN variants have been recently proposed to address this problem. They can be grouped into two main categories: training either a single generator or many generators. Methods in the former include modifying the discriminator's objective (Salimans et al., 2016; Metz et al., 2016), modifying the generator's objective (Warde-Farley & Bengio, 2016), or employing additional discriminators to yield more useful gradient signals for the generators (Nguyen et al., 2017; Durugkar et al., 2016). The common theme in these variants is that generators are shown, at equilibrium, to be able to recover the data distribution, but convergence remains elusive in practice. Most experiments are conducted on toy datasets or on narrow-domain datasets such as LSUN (Yu et al., 2015) or CelebA (Liu et al., 2015). To our knowledge, only Warde-Farley & Bengio (2016) and Nguyen et al. (2017) perform quantitative evaluation of models trained on much more diverse datasets such as STL-10 (Coates et al., 2011) and ImageNet (Russakovsky et al., 2015).

Given current limitations in the training of single-generator GANs, some very recent attempts have been made following the multi-generator approach. Tolstikhin et al. (2017) apply boosting techniques to train a mixture of generators by sequentially training and adding new generators to the mixture. However, sequentially training many generators is computational expensive. Moreover, this approach is built on the implicit assumption that a single-generator GAN can generate very good images of some modes, so reweighing the training data and incrementally training new generators will result in a mixture that covers the whole data space. This assumption is not true in practice since current single-generator GANs trained on diverse datasets such as ImageNet tend to generate images of unrecognizable objects. Arora et al. (2017) train a mixture of generators and discriminators, and optimize the minimax game with the reward function being the weighted average reward function between any pair of generator and discriminator. This model is computationally expensive and lacks a mechanism to enforce the divergence among generators. Ghosh et al. (2017) train many generators by using a multi-class discriminator that, in addition to detecting whether a data sample is fake, predicts which generator produces the sample. The objective function in this model punishes generators for generating samples that are detected as fake but does not directly encourage generators to specialize in generating different types of data.

We propose in this paper a novel approach to train a mixture of generators. Unlike aforementioned multi-generator GANs, our proposed model simultaneously trains a set of generators with the objective that the mixture of their induced distributions would approximate the data distribution, whilst encouraging them to specialize in different data modes. The result is a novel adversarial architecture formulated as a minimax game among three parties: a classifier, a discriminator, and a set of generators. Generators create samples that are intended to come from the same distribution as the training data, whilst the discriminator determines whether samples are true data or generated by generators, and the classifier specifies which generator a sample comes from. We term our proposed model as *Mixture Generative Adversarial Nets (MGAN)*. We provide analysis that our model is optimized towards minimizing the Jensen-Shannon Divergence (JSD) between the mixture of distributions induced by the generators and the data distribution while maximizing the JSD among generators.

Empirically, our proposed model can be trained efficiently by utilizing parameter sharing among generators, and between the classifier and the discriminator. In addition, simultaneously training many generators while enforcing JSD among generators helps each of them focus on some modes of the data space and learn better. Trained on CIFAR-10, each generator learned to specialize in generating samples from a different class such as horse, car, ship, dog, bird or airplane. Overall, the models trained on the CIFAR-10, STL-10 and ImageNet datasets successfully generated diverse, recognizable objects and achieved state-of-the-art Inception scores (Salimans et al., 2016). The model trained on the CIFAR-10 even outperformed GANs trained in a semi-supervised fashion (Salimans et al., 2016; Odena et al., 2016).

In short, our main contributions are: (i) a novel adversarial model to efficiently train a mixture of generators while enforcing the JSD among the generators; (ii) a theoretical analysis that our objective function is optimized towards minimizing the JSD between the mixture of all generators' distributions and the real data distribution, while maximizing the JSD among generators; and (iii) a comprehensive evaluation on the performance of our method on both synthetic and real-world large-scale datasets of diverse natural scenes.

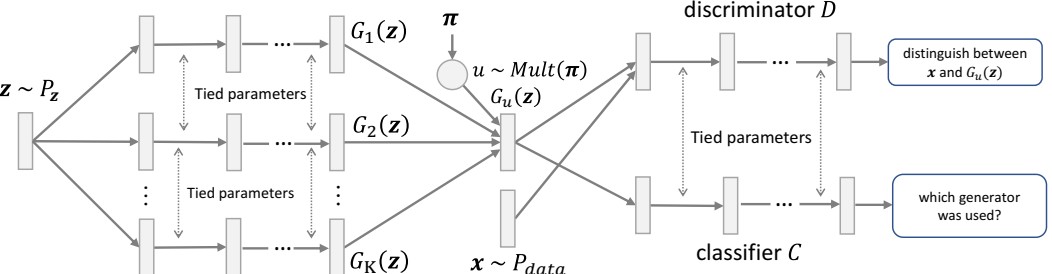

Figure 1: MGAN's architecture with K generators, a binary discriminator, a multi-class classifier.

## 2 GENERATIVE ADVERSARIAL NETS

Given the discriminator $D$ and generator $G$, both parameterized via neural networks, training GAN can be formulated as the following minimax objective function:

$$\min_{G} \max_{D} \mathbb{E}_{\mathbf{x} \sim P_{data}(\mathbf{x})} \left[ \log D(\mathbf{x}) \right] + \mathbb{E}_{\mathbf{z} \sim P_{\mathbf{z}}} \left[ \log \left( 1 - D(G(\mathbf{z})) \right) \right] \tag{1}$$

where $\mathbf{x}$ is drawn from data distribution $P_{data}$, $\mathbf{z}$ is drawn from a prior distribution $P_{\mathbf{z}}$. The mapping $G(\mathbf{z})$ induces a generator distribution $P_{model}$ in data space. GAN alternatively optimizes $D$ and $G$ using stochastic gradient-based learning. As a result, the optimization order in 1 can be reversed, causing the minimax formulation to become maximin. $G$ is therefore incentivized to map every $\mathbf{z}$ to a single $\mathbf{x}$ that is most likely to be classified as true data, leading to mode collapsing problem. Another commonly asserted cause of generating less diverse samples in GAN is that, at the optimal point of $D$, minimizing $G$ is equivalent to minimizing the JSD between the data and model distributions, which has been empirically proven to prefer to generate samples around only a few modes whilst ignoring other modes (Huszár, 2015; Theis et al., 2015).

## 3 PROPOSED MIXTURE GANS

We now present our main contribution of a novel approach that can effectively tackle mode collapse in GAN. Our idea is to use a mixture of many distributions rather than a single one as in the standard GAN, to approximate the data distribution, and simultaneously we enlarge the divergence of those distributions so that they cover different data modes.

To this end, an analogy to a game among K generators $G_{1:K}$, a discriminator $D$ and a classifier $C$ can be formulated. Each generator $G_k$ maps $\mathbf{z}$ to $\mathbf{x} = G_k(\mathbf{z})$, thus inducing a single distribution $P_{G_k}$; and K generators altogether induce a mixture over K distributions, namely $P_{model}$ in the data space. An index $u$ is drawn from a multinomial distribution $\mathrm{Mult}(\boldsymbol{\pi})$ where $\boldsymbol{\pi} = [\pi_1, \pi_2, ..., \pi_K]$ is the coefficients of the mixture; and then the sample $G_u(\mathbf{z})$ is used as the output. Here, we use a predefined $\boldsymbol{\pi}$ and fix it instead of learning. The discriminator $D$ aims to distinguish between this sample and the training samples. The classifier $C$ performs multi-class classification to classify samples labeled by the indices of their corresponding generators. We term this whole process and our model the *Mixture Generative Adversarial Nets* (MGAN).

Fig. 1 illustrates the general architecture of our proposed MGAN, where all components are parameterized by neural networks. $G_k$ (s) tie their parameters together except the input layer, whilst $C$ and $D$ share parameters except the output layer. This parameter sharing scheme enables the networks to leverage their common information such as features at low-level layers that are close to the data layer, hence helps to train model effectively. In addition, it also minimizes the number of parameters and adds minimal complexity to the standard GAN, thus the whole process is still very efficient.

More formally, $D$, $C$ and $G_{1:K}$ now play the following multi-player minimax optimization game:

$$\min_{G_{1:K}, C} \max_{D} \mathcal{J}(G_{1:K}, C, D) = \mathbb{E}_{\mathbf{x} \sim P_{data}} \left[ \log D(\mathbf{x}) \right] + \mathbb{E}_{\mathbf{x} \sim P_{model}} \left[ \log \left( 1 - D(\mathbf{x}) \right) \right]$$

$$- \beta \left\{ \sum_{k=1}^{K} \pi_k \mathbb{E}_{\mathbf{x} \sim P_{G_k}} \left[ \log C_k(\mathbf{x}) \right] \right\} \tag{2}$$

where $C_k(\mathbf{x})$ is the probability that $\mathbf{x}$ is generated by $G_k$ and $\beta > 0$ is the diversity hyper-parameter. The first two terms show the interaction between generators and the discriminator as in the standard GAN. The last term should be recognized as a standard softmax loss for a multi-classification setting, which aims to maximize the entropy for the classifier. This represents the interaction between generators and the classifier, which encourages each generator to produce data separable from those produced by other generators. The strength of this interaction is controlled by $\beta$. Similar to GAN, our proposed network can be trained by alternatively updating $D$, $C$ and $G_{1:K}$. We refer to Appendix A for the pseudo-code and algorithms for parameter learning for our proposed MGAN.

### 3.1 Theoretical Analysis

Assuming all $C$, $D$ and $G_{1:K}$ have enough capacity, we show below that at the equilibrium point of the minimax problem in Eq. (2), the JSD between the mixture induced by $G_{1:K}$ and the data distribution is minimal, i.e. $p_{data} = p_{model}$, and the JSD among K generators is maximal, i.e. two arbitrary generators almost never produce the same data. In what follows we present our mathematical statement and the sketch of their proofs. We refer to Appendix B for full derivations.

**Proposition 1.** *For fixed generators $G_1$, $G_2$, ..., $G_K$ and their mixture weights $\pi_1, \pi_2, ..., \pi_K$, the optimal solution $C^* = C_{1:K}^*$ and $D^*$ for $\mathcal{J}(G_{1:K}, C, D)$ in Eq. (2) are:*

$$C_k^*(\mathbf{x}) = \frac{\pi_k p_{G_k}(\mathbf{x})}{\sum_{j=1}^{K} \pi_j p_{G_j}(\mathbf{x})} \quad \text{and} \quad D^*(\mathbf{x}) = \frac{p_{data}(\mathbf{x})}{p_{data}(\mathbf{x}) + p_{model}(\mathbf{x})}$$

*Proof.* It can be seen that the solution $C_k^*$ is a general case of $D^*$ when $D$ classifies samples from two distributions with equal weight of $1/2$. We refer the proofs for $D^*$ to Prop. 1 in (Goodfellow et al., 2014), and our proof for $C_k^*$ to Appendix B in this manuscript. $\square$

Based on Prop. 1, we further show that at the equilibrium point of the minimax problem in Eq. (2), the optimal generator $G^* = [G_1^*, ..., G_K^*]$ induces the generated distribution $p_{model}^*(\mathbf{x}) = \sum_{k=1}^{K} \pi_k p_{G_k^*}(\mathbf{x})$ which is as closest as possible to the true data distribution $p_{data}(\mathbf{x})$ while maintaining the mixture components $p_{G_k^*}(\mathbf{x})$(s) as furthest as possible to avoid the mode collapse.

**Theorem 2.** *At the equilibrium point of the minimax problem in Eq. (2), the optimal $G^*, D^*$, and $C^*$ satisfy*

$$G^* = \underset{G}{argmin} \; (2 \cdot \text{JSD}(P_{data} \| P_{model}) - \beta \cdot \text{JSD}_{\boldsymbol{\pi}}(P_{G_1}, P_{G_2}, ..., P_{G_K})) \tag{3}$$

$$C_k^*(\mathbf{x}) = \frac{\pi_k p_{G_k^*}(\mathbf{x})}{\sum_{j=1}^{K} \pi_j p_{G_j^*}(\mathbf{x})} \quad \text{and} \quad D^*(\mathbf{x}) = \frac{p_{data}(\mathbf{x})}{p_{data}(\mathbf{x}) + p_{model}(\mathbf{x})}$$

*Proof.* Substituting $C_{1:K}^*$ and $D^*$ into Eq. (2), we reformulate the objective function for $G_{1:K}$ as follows:

$$\mathcal{L}(G_{1:K}) = \mathbb{E}_{\mathbf{x} \sim P_{data}}\left[\log \frac{p_{data}(\mathbf{x})}{p_{data}(\mathbf{x}) + p_{model}(\mathbf{x})}\right] + \mathbb{E}_{\mathbf{x} \sim P_{model}}\left[\log \frac{p_{model}(\mathbf{x})}{p_{data}(\mathbf{x}) + p_{model}(\mathbf{x})}\right]$$

$$- \beta \left\{\sum_{k=1}^{K} \pi_k \mathbb{E}_{\mathbf{x} \sim P_{G_k}}\left[\log \frac{\pi_k p_{G_k}(\mathbf{x})}{\sum_{j=1}^{K} \pi_j p_{G_j}(\mathbf{x})}\right]\right\}$$

$$= 2 \cdot \text{JSD}(P_{data} \| P_{model}) - \log 4 - \beta \left\{\sum_{k=1}^{K} \pi_k \mathbb{E}_{\mathbf{x} \sim P_{G_k}}\left[\log \frac{p_{G_k}(\mathbf{x})}{\sum_{j=1}^{K} \pi_j p_{G_j}(\mathbf{x})}\right]\right\} - \beta \sum_{k=1}^{K} \pi_k \log \pi_k$$

$$= 2 \cdot \text{JSD}(P_{data} \| P_{model}) - \beta \cdot \text{JSD}_{\boldsymbol{\pi}}(P_{G_1}, P_{G_2}, ..., P_{G_K}) - \log 4 - \beta \sum_{k=1}^{K} \pi_k \log \pi_k \tag{4}$$

Since the last two terms in Eq. (4) are constant, that concludes our proof. $\square$

This theorem shows that progressing towards the equilibrium is equivalently to minimizing $\text{JSD}(P_{data} \| P_{model})$ while maximizing $\text{JSD}_{\boldsymbol{\pi}}(P_{G_1}, P_{G_2}, ..., P_{G_K})$. In the next theorem, we further clarify the equilibrium point for the specific case wherein the data distribution has the form

$p_{data}(\mathbf{x}) = \sum_{k=1}^{K} \pi_k q_k(\mathbf{x})$ where the mixture components $q_k(\mathbf{x})$(s) are well-separated in the sense that $\mathbb{E}_{\mathbf{x} \sim Q_k}[q_j(\mathbf{x})] = 0$ for $j \neq k$, i.e., for almost everywhere $\mathbf{x}$, if $q_k(\mathbf{x}) > 0$ then $q_j(\mathbf{x}) = 0, \forall j \neq k$.

**Theorem 3.** *If the data distribution has the form:* $p_{data}(\mathbf{x}) = \sum_{k=1}^{K} \pi_k q_k(\mathbf{x})$ *where the mixture components* $q_k(\mathbf{x})$(s) *are well-separated, the minimax problem in Eq. (2) or the optimization problem in Eq. (3) has the following solution:*

$$p_{G_k^*}(\mathbf{x}) = q_k(\mathbf{x}), \, \forall k = 1, \ldots, K \text{ and } p_{model}(\mathbf{x}) = \sum_{k=1}^{K} \pi_k q_k(\mathbf{x}) = p_{data}(\mathbf{x})$$

*, and the corresponding objective value of the optimization problem in Eq. (3) is* $-\beta\mathbb{H}(\boldsymbol{\pi}) = -\beta \sum_{k=1}^{K} \pi_k \log \frac{1}{\pi_k}$, *where* $\mathbb{H}(\boldsymbol{\pi})$ *is the Shannon entropy.*

*Proof.* Please refer to our proof in Appendix B of this manuscript. $\square$

Thm. 3 explicitly offers the optimal solution for the specific case wherein the real data are generated from a mixture distribution whose components are well-separated. This further reveals that if the mixture components are well-separated, by setting the number of generators as the number of mixtures in data and maximizing the divergence between the generated components $p_{G_k}(\mathbf{x})$(s), we can exactly recover the mixture components $q_k(\mathbf{x})$(s) using the generated components $p_{G_k}(\mathbf{x})$(s), hence strongly supporting our motivation when developing MGAN. In practice, $C$, $D$, and $G_{1:K}$ are parameterized by neural networks and are optimized in the parameter space rather than in the function space. As all generators $G_{1:K}$ share the same objective function, we can efficiently update their weights using the same backpropagation passes. Empirically, we set the parameter $\pi_k = \frac{1}{K}, \forall k \in \{1, ..., K\}$, which further minimizes the objective value $-\beta\mathbb{H}(\boldsymbol{\pi}) = -\beta \sum_{k=1}^{K} \pi_k \log \frac{1}{\pi_k}$ w.r.t $\boldsymbol{\pi}$ in Thm. 3. To simplify the computational graph, we assume that each generator is sampled the same number of times in each minibatch. In addition, we adopt the non-saturating heuristic proposed in (Goodfellow et al., 2014) to train $G_{1:K}$ by maximizing $\log D(G_k(\mathbf{z}))$ instead of minimizing $\log D(1 - G_k(\mathbf{z}))$.

## 4 RELATED WORK

Recent attempts to address the mode collapse by modifying the discriminator include minibatch discrimination (Salimans et al., 2016), Unrolled GAN (Metz et al., 2016) and Denoising Feature Matching (DFM) (Warde-Farley & Bengio, 2016). The idea of minibatch discrimination is to allow the discriminator to detect samples that are noticeably similar to other generated samples. Although this method can generate visually appealing samples, it is computationally expensive, thus normally used in the last hidden layer of discriminator. Unrolled GAN improves the learning by unrolling computational graph to include additional optimization steps of the discriminator. It could effectively reduce the mode collapsing problem, but the unrolling step is expensive, rendering it unscalable up to large-scale datasets. DFM augments the objective function of generator with one of a Denoising AutoEncoder (DAE) that minimizes the reconstruction error of activations at the penultimate layer of the discriminator. The idea is that gradient signals from DAE can guide the generator towards producing samples whose activations are close to the manifold of real data activations. DFM is surprisingly effective at avoiding mode collapse, but the involvement of a deep DAE adds considerable computational cost to the model.

An alternative approach is to train additional discriminators. D2GAN (Nguyen et al., 2017) employs two discriminators to minimize both Kullback-Leibler (KL) and reverse KL divergences, thus placing a fair distribution across the data modes. This method can avoid the mode collapsing problem to a certain extent, but still could not outperform DFM. Another work uses many discriminators to boost the learning of generator (Durugkar et al., 2016). The authors state that this method is robust to mode collapse, but did not provide experimental results to support that claim.

Another direction is to train multiple generators. The so-called MIX+GAN (Arora et al., 2017) is related to our model in the use of mixture but the idea is very different. Based on min-max theorem (Neumann, 1928), the MIX+GAN trains a mixture of multiple generators and discriminators with

*different parameters* to play mixed strategies in a min-max game. The total reward of this game is computed by weighted averaging rewards over all pairs of generator and discriminator. The lack of parameter sharing renders this method computationally expensive to train. Moreover, there is no mechanism to enforce the divergence among generators as in ours.

Some attempts have been made to train a mixture of GANs in a similar spirit with boosting algorithms. Wang et al. (2016) propose an additive procedure to incrementally train new GANs on a subset of the training data that are badly modeled by previous generators. As the discriminator is expected to classify samples from this subset as real with high confidence, i.e. $D(\mathbf{x})$ is high, the subset can be chosen to include $\mathbf{x}$ where $D(\mathbf{x})$ is larger than a predefined threshold. Tolstikhin et al. (2017), however, show that this heuristic fails to address the mode collapsing problem. Thus they propose AdaGAN to introduce a robust reweighing scheme to prepare training data for the next GAN. AdaGAN and boosting-inspired GANs in general are based on the assumption that a single-generator GAN can learn to generate impressive images of some modes such as dogs or cats but fails to cover other modes such as giraffe. Therefore, removing images of dogs or cats from the training data and train a next GAN can create a better mixture. This assumption is not true in practice as current single-generator GANs trained on diverse data sets such as ImageNet (Russakovsky et al., 2015) tend to generate images of unrecognizable objects.

The most closely related to ours is MAD-GAN (Ghosh et al., 2017) which trains many generators and uses a multi-class classifier as the discriminator. In this work, two strategies are proposed to address the mode collapse: (i) augmenting generator's objective function with a user-defined similarity based function to encourage different generators to generate diverse samples, and (ii) modifying discriminator's objective functions to push different generators towards different identifiable modes by separating samples of each generator. Our approach is different in that, rather than modifying the discriminator, we use an additional classifier that discriminates samples produced by each generator from those by others under multi-class classification setting. This nicely results in an optimization problem that maximizes the JSD among generators, thus naturally enforcing them to generate diverse samples and effectively avoiding mode collapse.

## 5 EXPERIMENTS

In this section, we conduct experiments on both synthetic data and real-world large-scale datasets. The aim of using synthetic data is to visualize, examine and evaluate the learning behaviors of our proposed MGAN, whilst using real-world datasets to quantitatively demonstrate its efficacy and scalability of addressing the mode collapse in a much larger and wider data space. For fair comparison, we use experimental settings that are identical to previous work, and hence we quote the results from the latest state-of-the-art GAN-based models to compare with ours.

We use TensorFlow (Abadi et al., 2016) to implement our model, and the source code is available at: https://github.com/qhoangdl/MGAN. For all experiments, we use: (i) shared parameters among generators in all layers except for the weights from the input to the first hidden layer; (ii) shared parameters between discriminator and classifier in all layers except for the weights from the penultimate layer to the output; (iii) Adam optimizer (Kingma & Ba, 2014) with learning rate of 0.0002 and the first-order momentum of 0.5; (iv) minibatch size of 64 samples for training discriminators; (v) ReLU activations (Nair & Hinton, 2010) for generators; (vi) Leaky ReLU (Maas et al., 2013) with slope of 0.2 for discriminator and classifier; and (vii) weights randomly initialized from Gaussian distribution $\mathcal{N}(0, 0.02\boldsymbol{I})$ and zero biases. We refer to Appendix C for detailed model architectures and additional experimental results.

### 5.1 SYNTHETIC DATA

In the first experiment, following (Nguyen et al., 2017) we reuse the experimental design proposed in (Metz et al., 2016) to investigate how well our MGAN can explore and capture multiple data modes. The training data is sampled from a 2D mixture of 8 isotropic Gaussian distributions with a covariance matrix of $0.02\boldsymbol{I}$ and means arranged in a circle of zero centroid and radius of 2.0. Our purpose of using such small variance is to create low density regions and separate the modes.

We employ 8 generators, each with a simple architecture of an input layer with 256 noise units drawn from isotropic multivariate Gaussian distribution $\mathcal{N}(0, \boldsymbol{I})$, and two fully connected hidden

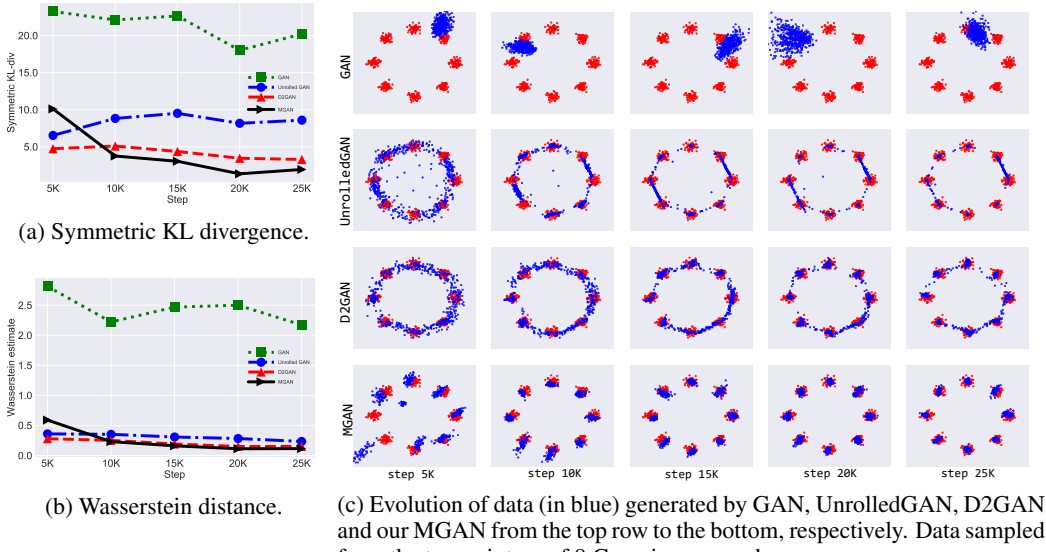

(a) Symmetric KL divergence.

(b) Wasserstein distance.

(c) Evolution of data (in blue) generated by GAN, UnrolledGAN, D2GAN and our MGAN from the top row to the bottom, respectively. Data sampled from the true mixture of 8 Gaussians are red.

Figure 2: The comparison of our MGAN and GAN's variants on 2D synthetic dataset.

layers with 128 ReLU units each. For the discriminator and classifier, one hidden layer with 128 ReLU units is used. The diversity hyperparameter $\beta$ is set to 0.125.

Fig. 2c shows the evolution of 512 samples generated by our model and baselines through time. It can be seen that the regular GAN generates data collapsing into a *single* mode hovering around the valid modes of data distribution, thus reflecting the mode collapse in GAN as expected. At the same time, UnrolledGAN (Metz et al., 2016), D2GAN (Nguyen et al., 2017) and our MGAN distribute data around *all* 8 mixture components, and hence demonstrating the abilities to successfully learn multimodal data in this case. Our proposed model, however, converges much faster than the other two since it successfully explores and neatly covers all modes at the early step 15K, whilst two baselines produce samples cycling around till the last steps. At the end, our MGAN captures data modes more precisely than UnrolledGAN and D2GAN since, in each mode, the UnrolledGAN generates data that concentrate only on several points around the mode's centroid, thus seems to produce fewer samples than ours whose samples fairly spread out the entire mode, but not exceed the boundary whilst the D2GAN still generates many points scattered between two adjacent modes.

Next we further quantitatively compare the quality of generated data. Since we know the true distribution $P_{data}$ in this case, we employ two measures, namely Wasserstein distance and symmetric Kullback-Leibler (KL) divergence, which is the average of KL and reverse KL. These measures compute the distance between the normalized histograms of 10,000 points generated from the model to true $P_{data}$. Figs. 2a and 2b again clearly demonstrate the superiority of our approach over GAN, UnrolledGAN and D2GAN w.r.t both distances (lower is better); notably the Wasserstein distances from ours and D2GAN's to the true distribution almost reduce to zero, and at the same time, our symmetric KL metric is significantly better than that of D2GAN. These figures also show the stability of our MGAN (black curves) and D2GAN (red curves) during training as they are much less fluctuating compared with GAN (green curves) and UnrolledGAN (blue curves).

Lastly, we perform experiments with different numbers of generators. The MGAN models with 2, 3, 4 and 10 generators all successfully explore 8 modes but the models with more generators generate fewer points scattered between adjacent modes. We also examine the behavior of the diversity coefficient $\beta$ by training the 4-generator model with different values of $\beta$. Without the JSD force ($\beta = 0$), generated samples cluster around one mode. When $\beta = 0.25$, the JSD force is weak and generated data cluster near 4 different modes. When $\beta = 0.75$ or $1.0$, the JSD force is too strong and causes the generators to collapse, generating 4 increasingly tight clusters. When $\beta = 0.5$, generators successfully cover all of the 8 modes. Please refer to Appendix C.1 for experimental details.

## 5.2 Real-world Datasets

Next we train our proposed method on real-world databases from natural scenes to investigate its performance and scalability on much more challenging large-scale image data.

**Datasets.** We use 3 widely-adopted datasets: CIFAR-10 (Krizhevsky & Hinton, 2009), STL-10 (Coates et al., 2011) and ImageNet (Russakovsky et al., 2015). CIFAR-10 contains 50,000 $32{\times}32$ training images of 10 classes: airplane, automobile, bird, cat, deer, dog, frog, horse, ship, and truck. STL-10, subsampled from ImageNet, is a more diverse dataset than CIFAR-10, containing about 100,000 $96{\times}96$ images. ImageNet (2012 release) presents the largest and most diverse consisting of over 1.2 million images from 1,000 classes. In order to facilitate fair comparison with the baselines in (Warde-Farley & Bengio, 2016; Nguyen et al., 2017), we follow the procedure of (Krizhevsky et al., 2012) to resize the STL-10 and ImageNet images down to $48{\times}48$ and $32{\times}32$, respectively.

**Evaluation protocols.** For quantitative evaluation, we adopt the Inception score proposed in (Salimans et al., 2016), which computes $\exp\left(\mathbb{E}_{\mathbf{x}}\left[KL\left(p\left(y|\mathbf{x}\right)\|p\left(y\right)\right)\right]\right)$ where $p\left(y|\mathbf{x}\right)$ is the conditional label distribution for the image $\mathbf{x}$ estimated by the reference Inception model (Szegedy et al., 2015). This metric rewards good and varied samples and is found to be well-correlated with human judgment (Salimans et al., 2016). We use the code provided in (Salimans et al., 2016) to compute the Inception scores for 10 partitions of 50,000 randomly generated samples. For qualitative demonstration of image quality obtained by our proposed model, we show samples generated by the mixture as well as samples produced by each generator. Samples are randomly drawn rather than cherry-picked.

**Model architectures.** Our generator and discriminator architectures closely follow the DCGAN's design (Radford et al., 2015). The only difference is we apply batch normalization (Ioffe & Szegedy, 2015) to all layers in the networks except for the output layer. Regarding the classifier, we empirically find that our proposed MGAN achieves the best performance (i.e., fast convergence rate and high inception score) when the classifier shares parameters of all layers with the discriminator except for the output layer. The reason is that this parameter sharing scheme would allow the classifier and discriminator to leverage their common features and representations learned at every layer, thus helps to improve and speed up the training progress. When the parameters are not tied, the model learns slowly and eventually yields lower performance.

During training we observe that the percentage of active neurons chronically declined (see Appendix C.2). One possible cause is that the batch normalization center (offset) is gradually shifted to the negative range, thus deactivating up to 45% of ReLU units of the generator networks. Our ad-hoc solution for this problem is to fix the offset at zero for all layers in the generator networks. The rationale is that for each feature map, the ReLU gates will open for about 50% highest inputs in a minibatch across all locations and generators, and close for the rest.

We also experiment with other activation functions of generator networks. First we use Leaky ReLU and obtain similar results with using ReLU. Then we use MaxOut units (Goodfellow et al., 2013) and achieves good Inception scores but generates unrecognizable samples. Finally, we try SeLU (Klambauer et al., 2017) but fail to train our model.

**Hyperparameters.** Three key hyperparameters of our model are: number of generators K, coefficient $\beta$ controlling the diversity and the minibatch size. We use a minibatch size of $[^{128}/_{\text{K}}]$ for each generator, so that the total number of samples for training all generators is about 128. We train models with 4 generators and 10 generators corresponding with minibatch sizes of 32 and 12 each, and find that models with 10 generators performs better. For ImageNet, we try an additional setting with 32 generators and a minibatch size of 4 for each. The batch of 4 samples is too small for updating sufficient statistics of a batch-norm layer, thus we drop batch-norm in the input layer of each generator. This 32-generator model, however, does not obtain considerably better results than the 10-generator one. Therefore in what follows we only report the results of models with 10 generators. For the diversity coefficient $\beta$, we observe no significant difference in Inception scores when varying the value of $\beta$ but the quality of generated images declines when $\beta$ is too low or too high. Generated samples by each generator vary more when $\beta$ is low, and vary less but become less realistic when $\beta$ is high. We find a reasonable range for $\beta$ to be $(0.01, 1.0)$, and finally set to 0.01 for CIFAR-10, 0.1 for ImageNet and 1.0 for STL-10.

**Inception results.** We now report the Inception scores obtained by our MGAN and baselines in Tab. 1. It is worthy to note that only models trained in a completely unsupervised manner *without label* information are included for fair comparison; and DCGAN's and D2GAN's results on STL-10 are available only for the models trained on 32×32 resolution. Overall, our proposed model outperforms the baselines by large margins and achieves state-of-the-art performance on all datasets. Moreover, we would highlight that our MGAN obtains a score of 8.33 on CIFAR-10 that is even better than those of models trained *with labels* such as 8.09 of Improved GAN (Salimans et al., 2016) and 8.25 of AC-GAN (Odena et al., 2016). In addition, we train our model on the original 96×96 resolution of STL-10 and achieve a score of 9.79±0.08. This suggests the MGAN can be successfully trained on higher resolution images and achieve the higher Inception score.

Table 1: Inception scores on different datasets. All models are trained in an unsupervised manner. "–" denotes unavailable result.

| Model | CIFAR-10 | STL-10 | ImageNet |
|---|---|---|---|
| Real data | 11.24±0.16 | 26.08±0.26 | 25.78±0.47 |
| WGAN (Arjovsky et al., 2017) | 3.82±0.06 | – | – |
| MIX+WGAN (Arora et al., 2017) | 4.04±0.07 | – | – |
| Improved-GAN (Salimans et al., 2016) | 4.36±0.04 | – | – |
| ALI (Dumoulin et al., 2016) | 5.34±0.05 | – | – |
| BEGAN (Berthelot et al., 2017) | 5.62 | – | – |
| MAGAN (Wang et al., 2017) | 5.67 | – | – |
| GMAN (Durugkar et al., 2016) | 6.00±0.19 | – | – |
| DCGAN (Radford et al., 2015) | 6.40±0.05 | 7.54 | 7.89 |
| DFM (Warde-Farley & Bengio, 2016) | 7.72±0.13 | 8.51±0.13 | 9.18±0.13 |
| D2GAN (Nguyen et al., 2017) | 7.15±0.07 | 7.98 | 8.25 |
| **MGAN** | **8.33±0.10** | **9.22±0.11** | **9.32±0.10** |

**Fréchet Inception Distance results.** One disadvantage of the Inception score is that it does not compare the statistics of real world samples and those of synthetic examples. Therefore, we further evaluate MGAN using the *Fréchet Inception Distance* (FID) proposed in (Heusel et al., 2017). Let $p$ and $q$ be the distributions of the representations obtained by projecting real and synthetic samples to the last hidden layer in Inception model (Szegedy et al., 2015). Assuming that $p$ and $q$ are both multivariate Gaussian distributions, FID measures the *Fréchet distance* (Dowson & Landau, 1982), which is also the 2-Wasserstein distance, between the two distributions. Tab. 2 compares the FIDs obtained by MGAN with baselines collected in (Heusel et al., 2017). It is noteworthy that lower FID is better, and that WGAN-GP and WGAN-GP + TTUR uses the ResNet architecture while MGAN employs the DCGAN architecture. In terms of FID, MGAN is roughly 28% better than DCGAN and DCGAN + TTUR, 9% better than WGAN-GP and 8% weaker than WGAN-GP + TTUR. This result further proves that MGAN helps address the mode collapsing problem.

Table 2: FIDs (lower is better) on CIFAR-10.

| Model | FID |
|---|---|
| DCGAN (Radford et al., 2015) | 37.7 |
| DCGAN + TTUR (Heusel et al., 2017) | 36.9 |
| WGAN-GP (Gulrajani et al., 2017) | 29.3 |
| WGAN-GP + TTUR (Heusel et al., 2017) | 24.8 |
| **MGAN** | **26.7** |

**Image generation.** Next we present samples randomly generated by our proposed model trained on the 3 datasets for qualitative assessment. Fig. 3a shows CIFAR-10 32×32 images containing a wide range of objects in such as airplanes, cars, trucks, ships, birds, horses or dogs. Similarly, STL-10 48×48 generated images in Fig. 3b include cars, ships, airplanes and many types of animals, but with wider range of different themes such as sky, underwater, mountain and forest. Images generated for ImageNet 32×32 are diverse with some recognizable objects such as lady, old man, birds, human eye, living room, hat, slippers, to name a few. Fig. 4a shows several cherry-picked STL-10 96×96

images, which demonstrate that the MGAN is capable of generating visually appealing images with complicated details. However, many samples are still incomplete and unrealistic as shown in Fig. 4b, leaving plenty of room for improvement.

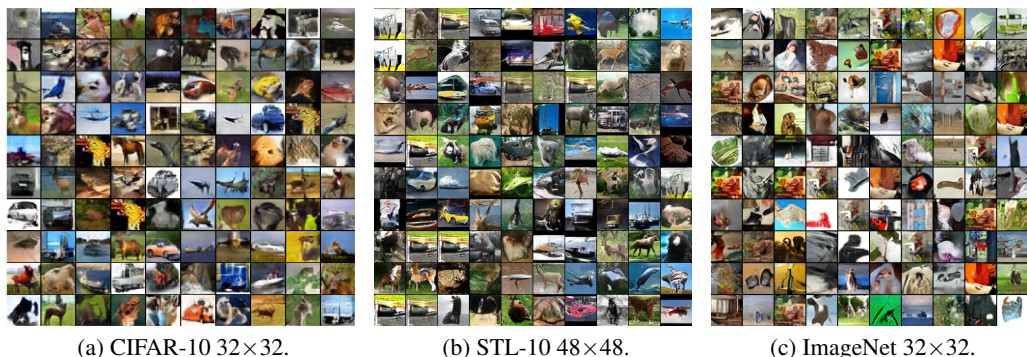

(a) CIFAR-10 32×32.          (b) STL-10 48×48.          (c) ImageNet 32×32.

Figure 3: Images generated by our proposed MGAN trained on natural image datasets. Due to the space limit, please refer to the appendix for larger plots.

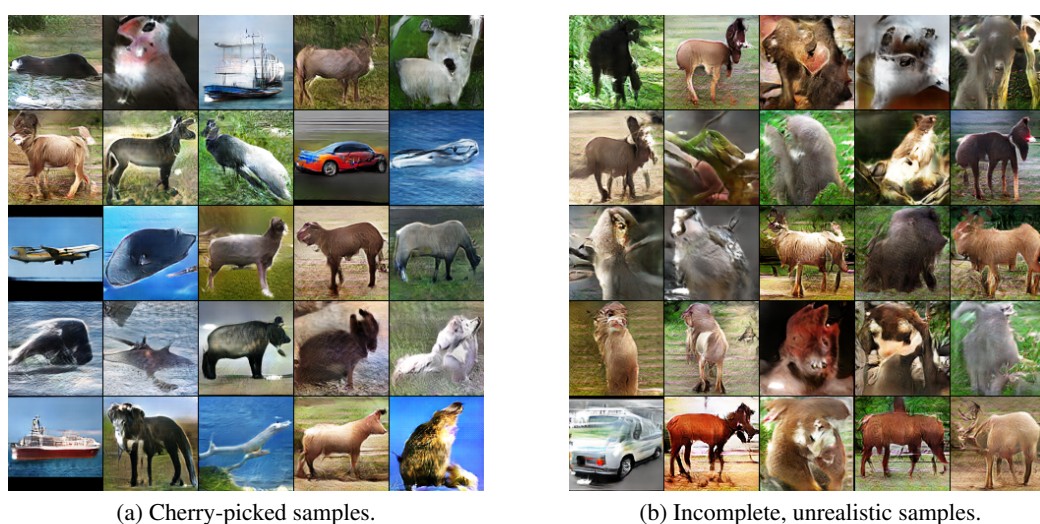

(a) Cherry-picked samples.          (b) Incomplete, unrealistic samples.

Figure 4: Images generated by our MGAN trained on the original 96×96 STL10 dataset.

Finally, we investigate samples generated by each generator as well as the evolution of these samples through numbers of training epochs. Fig. 5 shows images generated by each of the 10 generators in our MGAN trained on CIFAR-10 at epoch 20, 50, and 250 of training. Samples in each row correspond to a different generator. Generators start to specialize in generating different types of objects as early as epoch 20 and become more and more consistent: generator 2 and 3 in flying objects (birds and airplanes), generator 4 in full pictures of cats and dogs, generator 5 in portraits of cats and dogs, generator 8 in ships, generator 9 in car and trucks, and generator 10 in horses. Generator 6 seems to generate images of frog or animals in a bush. Generator 7, however, collapses in epoch 250. One possible explanation for this behavior is that images of different object classes tend to have different themes. Lastly, Wang et al. (2016) noticed one of the causes for non-convergence in GANs is that the generators and discriminators constantly vary; the generators at two consecutive epochs of training generate significantly different images. This experiment demonstrates the effect of the JSD force in preventing generators from moving around the data space.

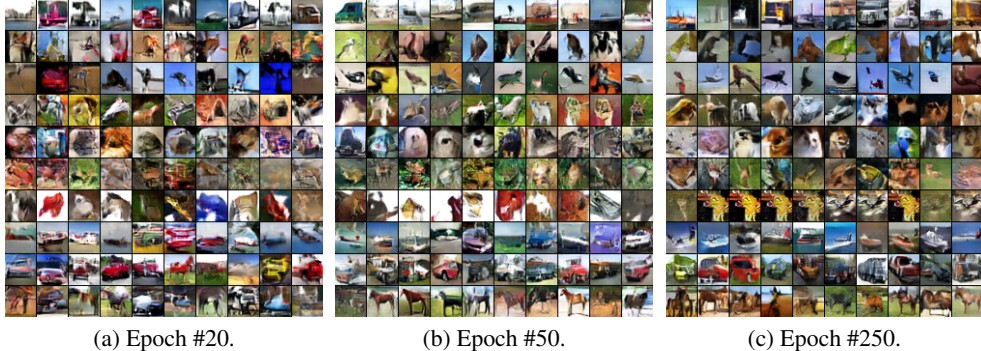

(a) Epoch #20.  (b) Epoch #50.  (c) Epoch #250.

Figure 5: Images generated by our MGAN trained on CIFAR10 at different epochs. Samples in each row from the top to the bottom correspond to a different generator.

# 6 CONCLUSION

We have presented a novel adversarial model to address the mode collapse in GANs. Our idea is to approximate data distribution using a mixture of multiple distributions wherein each distribution captures a subset of data modes separately from those of others. To achieve this goal, we propose a minimax game of one discriminator, one classifier and many generators to formulate an optimization problem that minimizes the JSD between $P_{data}$ and $P_{model}$, i.e., a mixture of distributions induced by the generators, whilst maximizes JSD among such generator distributions. This helps our model generate diverse images to better cover data modes, thus effectively avoids mode collapse. We term our proposed model *Mixture Generative Adversarial Network* (MGAN).

The MGAN can be efficiently trained by sharing parameters between its discriminator and classifier, and among its generators, thus our model is scalable to be evaluated on real-world large-scale datasets. Comprehensive experiments on synthetic 2D data, CIFAR-10, STL-10 and ImageNet databases demonstrate the following capabilities of our model: (i) achieving state-of-the-art Inception scores; (ii) generating diverse and appealing recognizable objects at different resolutions; and (iv) specializing in capturing different types of objects by the generators.

**Acknowledgments.** This work was partially supported by the Australian Research Council (ARC) DP160109394 and AOARD (FA2386-16-1-4138)

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

## A  APPENDIX: FRAMEWORK

In our proposed method, generators $G_1$, $G_2$, ... $G_K$ are deep convolutional neural networks parameterized by $\boldsymbol{\theta}_G$. These networks share parameters in all layers except for the input layers. The input layer for generator $G_k$ is parameterized by the mapping $f_{\boldsymbol{\theta}_G,k}(\mathbf{z})$ that maps the sampled noise $\mathbf{z}$ to the first hidden layer activation $\mathbf{h}$. The shared layers are parameterized by the mapping $g_{\boldsymbol{\theta}_G}(\mathbf{h})$ that maps the first hidden layer to the generated data. The pseudo-code of sampling from the mixture is described in Alg. 1. Classifier $C$ and classifier $D$ are also deep convolutional neural networks that are both parameterized by $\boldsymbol{\theta}_{CD}$. They share parameters in all layers except for the last layer. The pseudo-code of alternatively learning $\boldsymbol{\theta}_G$ and $\boldsymbol{\theta}_{CD}$ using stochastic gradient descend is described in Alg. 2.

---

**Algorithm 1** Sampling from MGAN's mixture of generators.

1: Sample noise $\mathbf{z}$ from the prior $P_{\mathbf{z}}$.
2: Sample a generator index $u$ from $\text{Mult}(\pi_1, \pi_2, ..., \pi_K)$ with predefined mixing probability $\boldsymbol{\pi} = (\pi_1, \pi_2, ..., \pi_K)$.
3: $\mathbf{h} = f_{\boldsymbol{\theta}_G, u}(\mathbf{z})$
4: $\mathbf{x} = g_{\boldsymbol{\theta}_G}(\mathbf{h})$
5: Return generated data $\mathbf{x}$ and the index $u$.

---

**Algorithm 2** Alternative training of MGAN using stochastic gradient descent.

1: **for** number of training iterations **do**
2:     Sample a minibatch of M data points $\left(\mathbf{x}^{(1)}, \mathbf{x}^{(2)}, ..., \mathbf{x}^{(M)}\right)$ from the data distribution $P_{data}$.
3:     Sample a minibatch of N generated data points $\left(\mathbf{x}'^{(1)}, \mathbf{x}'^{(2)}, ..., \mathbf{x}'^{(N)}\right)$ and N indices $(u_1, u_2, ..., u_N)$ from the current mixture.
4:     $\mathcal{L}_C = -\frac{1}{N} \sum_{n=1}^{N} \log C_{u_n}\left(\mathbf{x}'^{(n)}\right)$
5:     $\mathcal{L}_D = -\frac{1}{M} \sum_{m=1}^{M} \log D\left(\mathbf{x}^{(m)}\right) - \frac{1}{N} \sum_{n=1}^{N} \log\left[1 - D\left(\mathbf{x}'^{(n)}\right)\right]$
6:     Update classifier $C$ and discriminator $D$ by descending along their gradient: $\nabla_{\boldsymbol{\theta}_{CD}}(\mathcal{L}_C + \mathcal{L}_D)$.
7:     Sample a minibatch of N generated data points $\left(\mathbf{x}'^{(1)}, \mathbf{x}'^{(2)}, ..., \mathbf{x}'^{(N)}\right)$ and N indices $(u_1, u_2, ..., u_N)$ from the current mixture.
8:     $\mathcal{L}_G = -\frac{1}{N} \sum_{n=1}^{N} \log D\left(\mathbf{x}'^{(n)}\right) - \frac{\beta}{N} \sum_{n=1}^{N} \log C_{u_n}\left(\mathbf{x}'^{(n)}\right)$
9:     Update the mixture of generators $G$ by ascending along its gradient: $\nabla_{\boldsymbol{\theta}_G}\mathcal{L}_G$.
10: **end for**

---

## B APPENDIX: PROOFS FOR SECTION 3.1

**Proposition 1 (Prop. 1 restated).** *For fixed generators $G_1$, $G_2$, ..., $G_K$ and mixture weights $\pi_1, \pi_2, ..., \pi_K$, the optimal classifier $C^* = C^*_{1:K}$ and discriminator $D^*$ for $\mathcal{J}(G, C, D)$ are:*

$$C_k^*(\mathbf{x}) = \frac{\pi_k p_{G_k}(\mathbf{x})}{\sum_{j=1}^{K} \pi_j p_{G_j}(\mathbf{x})}$$

$$D^*(\mathbf{x}) = \frac{p_{data}(\mathbf{x})}{p_{data}(\mathbf{x}) + p_{model}(\mathbf{x})}$$

*Proof.* The optimal $D^*$ was proved in Prop. 1 in (Goodfellow, 2016). This section shows a similar proof for the optimal $C^*$. Assuming that $C^*$ can be optimized in the functional space, we can calculate the functional derivatives of $\mathcal{J}(G, C, D)$ with respect to each $C_k(\mathbf{x})$ for $k \in \{2, ..., K\}$ and set them equal to zero:

$$\frac{\delta \mathcal{J}}{\delta C_k(\mathbf{x})} = -\beta \frac{\delta}{\delta C_k(\mathbf{x})} \int \left(\pi_1 p_{G_1}(\mathbf{x}) \log\left(1 - \sum_{k=2}^{K} C_k(\mathbf{x})\right) + \sum_{k=2}^{K} \pi_k p_{G_k}(\mathbf{x}) \log C_k(\mathbf{x})\right) \mathrm{d}x$$

$$= -\beta \left(\frac{\pi_k p_{G_k}(\mathbf{x})}{C_k(\mathbf{x})} - \frac{\pi_1 p_{G_1}(\mathbf{x})}{C_1(\mathbf{x})}\right) \tag{5}$$

Setting $\frac{\delta \mathcal{J}(G,C,D)}{\delta C_k(\mathbf{x})}$ to 0 for $k \in \{2, ..., K\}$, we get:

$$\frac{\pi_1 p_{G_1}(\mathbf{x})}{C_1^*(\mathbf{x})} = \frac{\pi_2 p_{G_2}(\mathbf{x})}{C_2^*(\mathbf{x})} = ... = \frac{\pi_K p_{G_K}(\mathbf{x})}{C_K^*(\mathbf{x})} \tag{6}$$

$C_k^*(\mathbf{x}) = \frac{\pi_k p_{G_k}(\mathbf{x})}{\sum_{j=1}^{K} \pi_j p_{G_j}(\mathbf{x})}$ results from Eq. (6) due to the fact that $\sum_{k=1}^{K} C_k^*(\mathbf{x}) = 1$. □

**Reformulation of $\mathcal{L}(G_{1:K})$.** Replacing the optimal $C^*$ and $D^*$ into Eq. (2), we can reformulate the objective function for the generator as follows:

$$\mathcal{L}(G_{1:K}) = \mathcal{J}(G, C^*, D^*)$$

$$= \mathbb{E}_{\mathbf{x} \sim P_{data}} \left[ \log \frac{p_{data}(\mathbf{x})}{p_{data}(\mathbf{x}) + p_{model}(\mathbf{x})} \right] + \mathbb{E}_{\mathbf{x} \sim P_{model}} \left[ \log \frac{p_{model}(\mathbf{x})}{p_{data}(\mathbf{x}) + p_{model}(\mathbf{x})} \right]$$

$$- \beta \left\{ \sum_{k=1}^{K} \pi_k \mathbb{E}_{\mathbf{x} \sim P_{G_k}} \left[ \log \frac{\pi_k p_{G_k}(\mathbf{x})}{\sum_{j=1}^{K} \pi_j p_{G_j}(\mathbf{x})} \right] \right\} \tag{7}$$

The sum of the first two terms in Eq. (7) was shown in (Goodfellow et al., 2014) to be $2 \cdot$ JSD $(P_{data} \| P_{model}) - \log 4$. The last term $\beta\{*\}$ of Eq. (7) is related to the JSD for the K distributions:

$$* = \sum_{k=1}^{K} \pi_k \mathbb{E}_{\mathbf{x} \sim P_{G_k}} \left[ \log \frac{\pi_k p_{G_k}(\mathbf{x})}{\sum_{j=1}^{K} \pi_j p_{G_j}(\mathbf{x})} \right]$$

$$= \sum_{k=1}^{K} \pi_k \mathbb{E}_{\mathbf{x} \sim P_{G_k}} [\log p_{G_k}(\mathbf{x})] - \sum_{k=1}^{K} \pi_k \mathbb{E}_{\mathbf{x} \sim P_{G_k}} \left[ \log \sum_{j=1}^{K} \pi_j p_{G_j}(\mathbf{x}) \right] + \sum_{k=1}^{K} \pi_k \log \pi_k$$

$$= - \sum_{k=1}^{K} \pi_k \mathbb{H}(p_{G_k}) + \mathbb{H} \left( \sum_{j=1}^{K} \pi_j p_{G_j}(\mathbf{x}) \right) + \sum_{k=1}^{K} \pi_k \log \pi_k$$

$$= \text{JSD}_{\boldsymbol{\pi}}(P_{G_1}, P_{G_2}, ..., P_{G_K}) + \sum_{k=1}^{K} \pi_k \log \pi_k \tag{8}$$

where $\mathbb{H}(P)$ is the Shannon entropy for distribution $P$. Thus, $\mathcal{L}(G_{1:K})$ can be rewritten as:

$$\mathcal{L}(G_{1:K}) = -\log 4 + 2 \cdot \text{JSD}(P_{data} \| P_{model}) - \beta \cdot \text{JSD}_{\boldsymbol{\pi}}(P_{G_1}, P_{G_2}, ..., P_{G_K}) - \beta \sum_{k=1}^{K} \pi_k \log \pi_k$$

**Theorem 3 (Thm. 3 restated).** *If the data distribution has the form:* $p_{data}(\mathbf{x}) = \sum_{k=1}^{K} \pi_k q_k(\mathbf{x})$ *where the mixture components* $q_k(\mathbf{x})(s)$ *are well-separated, the minimax problem in Eq. (2) or the optimization problem in Eq. (3) has the following solution:*

$$p_{G_k^*}(\mathbf{x}) = q_k(\mathbf{x}), \forall k = 1, \dots, K \text{ and } p_{model}(\mathbf{x}) = \sum_{k=1}^{K} \pi_k q_k(\mathbf{x}) = p_{data}(\mathbf{x})$$

*, and the corresponding objective value of the optimization problem in Eq. (3) is* $-\beta \mathbb{H}(\boldsymbol{\pi}) = -\beta \sum_{k=1}^{K} \pi_k \log \frac{1}{\pi_k}$.

*Proof.* We first recap the optimization problem for finding the optimal $G^*$:

$$\min_{G} \left( 2 \cdot \text{JSD}(P_{data} \| P_{model}) - \beta \cdot \text{JSD}_{\boldsymbol{\pi}}(P_{G_1}, P_{G_2}, ..., P_{G_K}) \right)$$

The JSD in Eq. (8) is given by:

$$\text{JSD}_{\boldsymbol{\pi}}(P_{G_1}, P_{G_2}, ..., P_{G_K}) = \sum_{k=1}^{K} \pi_k \mathbb{E}_{\mathbf{x} \sim P_{G_k}} \left[ \log \frac{\pi_k p_{G_k}(\mathbf{x})}{\sum_{j=1}^{K} \pi_j p_{G_j}(\mathbf{x})} \right] - \sum_{k=1}^{K} \pi_k \log \pi_k \tag{9}$$

The $i$-th expectation in Eq. (9) can be derived as follows:

$$\mathbb{E}_{\mathbf{x} \sim P_{G_k}} \left[ \log \frac{\pi_k p_{G_k}(\mathbf{x})}{\sum_{j=1}^{K} \pi_j p_{G_j}(\mathbf{x})} \right] \leq \mathbb{E}_{\mathbf{x} \sim P_{G_k}} [\log 1] \leq 0$$

and the equality occurs if $\frac{\pi_k p_{G_k}(\mathbf{x})}{\sum_{j=1}^{K} \pi_j p_{G_j}(\mathbf{x})} = 1$ almost everywhere or equivalently for almost every $\mathbf{x}$ except for those in a zero measure set, we have:

$$p_{G_k}(\mathbf{x}) > 0 \implies p_{G_j}(\mathbf{x}) = 0, \forall j \neq k \tag{10}$$

Therefore, we obtain the following inequality:

$$\mathrm{JSD}_{\boldsymbol{\pi}}\left(P_{G_1}, P_{G_2}, ..., P_{G_\mathrm{K}}\right) \leq -\sum_{k=1}^{\mathrm{K}} \pi_k \log \pi_k = \sum_{k=1}^{\mathrm{K}} \pi_k \log \frac{1}{\pi_k} = \mathbb{H}\left(\boldsymbol{\pi}\right)$$

and the equality occurs if for almost every $\mathbf{x}$ except for those in a zero measure set, we have:

$$\forall k : p_{G_k}\left(\mathbf{x}\right) > 0 \Longrightarrow p_{G_j}\left(\mathbf{x}\right) = 0, \forall j \neq k$$

It follows that

$$2 \cdot \mathrm{JSD}\left(P_{data} \| P_{model}\right) - \beta \cdot \mathrm{JSD}_{\boldsymbol{\pi}}\left(P_{G_1}, P_{G_2}, ..., P_{G_\mathrm{K}}\right) \geq 0 - \beta \mathbb{H}\left(\boldsymbol{\pi}\right) = -\beta \mathbb{H}\left(\boldsymbol{\pi}\right)$$

and we peak the minimum if $p_{G_k} = q_k, \forall k$ since this solution satisfies both

$$p_{model}\left(\mathbf{x}\right) = \sum_{k=1}^{\mathrm{K}} \pi_k q_k\left(\mathbf{x}\right) = p_{data}\left(\mathbf{x}\right)$$

and the conditions depicted in Eq. (10). That concludes our proof. □

## C APPENDIX: ADDITIONAL EXPERIMENTS

### C.1 SYNTHETIC 2D GAUSSIAN DATA

The true data is sampled from a 2D mixture of 8 Gaussian distributions with a covariance matrix $0.02\boldsymbol{I}$ and means arranged in a circle of zero centroid and radius 2.0. We use a simple architecture of 8 generators with two fully connected hidden layers and a classifier and a discriminator with one shared hidden layer. All hidden layers contain the same number of 128 ReLU units. The input layer of generators contains 256 noise units sampled from isotropic multivariate Gaussian distribution $\mathcal{N}\left(0, \boldsymbol{I}\right)$. We do not use batch normalization in any layer. We refer to Tab. 3 for more specifications of the network and hyperparameters. "Shared" is short for parameter sharing among generators or between the classifier and the discriminator. Feature maps of 8/1 in the last layer for $C$ and $D$ means that two separate fully connected layers are applied to the penultimate layer, one for $C$ that outputs 8 logits and another for $D$ that outputs 1 logit.

Table 3: Network architecture and hyperparameters for 2D Gaussian data.

| Operation | Feature maps | Nonlinearity | Shared? |
|---|---|---|---|
| $G\left(\mathbf{z}\right) : \mathbf{z} \sim \mathcal{N}\left(0, \mathbf{I}\right)$ | 256 | | |
| Fully connected | 128 | ReLU | × |
| Fully connected | 128 | ReLU | √ |
| Fully connected | 2 | Linear | √ |
| $C\left(\mathbf{x}\right), D\left(\mathbf{x}\right)$ | 2 | | |
| Fully connected | 128 | Leaky ReLU | √ |
| Fully connected | 8/1 | Softmax/Sigmoid | × |
| Number of generators | 8 | | |
| Batch size for real data | 512 | | |
| Batch size for each generator | 128 | | |
| Number of iterations | 25,000 | | |
| Leaky ReLU slope | 0.2 | | |
| Learning rate | 0.0002 | | |
| Regularization constants | $\beta = 0.125$ | | |
| Optimizer | Adam($\beta_1 = 0.5, \beta_2 = 0.999$) | | |
| Weight, bias initialization | $\mathcal{N}\left(\mu = 0, \sigma = 0.02\boldsymbol{I}\right), 0$ | | |

**The effect of the number of generators on generated samples.** Fig. 6 shows samples produced by MGANs with different numbers of generators trained on synthetic data for 25,000 epochs. The model with 1 generator behaves similarly to the standard GAN as expected. The models with 2, 3 and 4 generators all successfully cover 8 modes, but the ones with more generators draw fewer points scattered between adjacent modes. Finally, the model with 10 generators also covers 8 modes wherein 2 generators share one mode and one generator hovering around another mode.

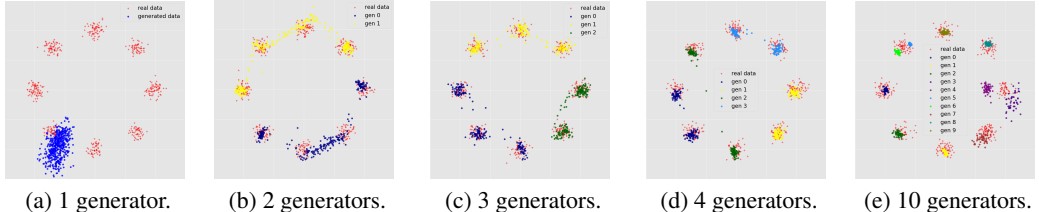

| (a) 1 generator. | (b) 2 generators. | (c) 3 generators. | (d) 4 generators. | (e) 10 generators. |

Figure 6: Samples generated by MGAN models trained on synthetic data with 2, 3, 4 and 10 generators. Data samples from the 8 Gaussians are in red, and generated data by each generator are in a different color.

**The effect of $\beta$ on generated samples.** To examine the behavior of the diversity coefficient $\beta$, Fig. 7 compares samples produced by our MGAN with 4 generators after 25,000 epochs of training with different values of $\beta$. Without the JSD force ($\beta = 0$), generated samples cluster around one mode. When $\beta = 0.25$, generated data clusters near 4 different modes. When $\beta = 0.75$ or $1.0$, the JSD force is too strong and causes the generators to collapse, generating 4 increasingly tight clusters. When $\beta = 0.5$, generators successfully cover all of the 8 modes.

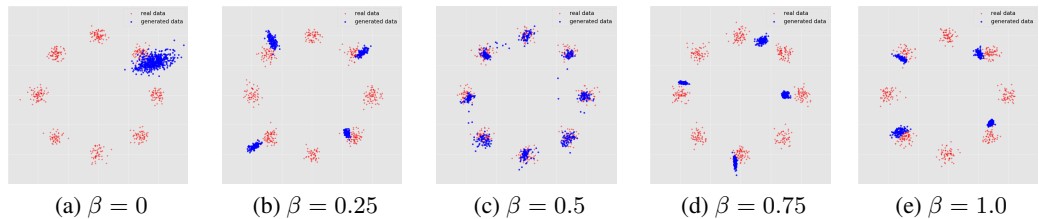

| (a) $\beta = 0$ | (b) $\beta = 0.25$ | (c) $\beta = 0.5$ | (d) $\beta = 0.75$ | (e) $\beta = 1.0$ |

Figure 7: Samples generated by MGAN models trained on synthetic data with different values of diversity coefficient $\beta$. Generated data are in blue and data samples from the 8 Gaussians are in red.

## C.2    REAL-WORLD DATASETS

**Fixing batch normalization center.** During training we observe that the percentage of active neurons, which we define as ReLU units with positive activation for at least 10% of samples in the minibatch, chronically declined. Fig. 8a shows the percentage of active neurons in generators trained on CIFAR-10 declined consistently to 55% in layer 2 and 60% in layer 3. Therefore, the quality of generated images, after reaching the peak level, started declining. One possible cause is that the batch normalization center (offset) is gradually shifted to the negative range as shown in the histogram in Fig. 8b. We also observe the same problem in DCGAN. Our ad-hoc solution for this problem, i.e., we fix the offset at zero for all layers in the generator networks. The rationale is that for each feature map, the ReLU gates will open for about 50% highest inputs in a minibatch across all locations and generators, and close for the rest. Therefore, batch normalization can keep ReLU units alive even when most of their inputs are otherwise negative, and introduces a form of competition that encourages generators to "specialize" in different features. This measure significantly improves performance but does not totally solve the dying ReLUs problem. We find that late in the training, the input to generators' ReLU units became more and more right-skewed, causing the ReLU gates to open less and less often.

**Parameter sharing.** We conduct experiment on CIFAR-10 without parameter sharing among generators. Surprisingly, 4 generators, each with 128 feature maps in the penultimate layer, fail to learn even when beta is set to 0.0. When the number of feature maps in the penultimate layer of each generator is set to 32, the model achieved an Inception Score of 7.42. Therefore, we hypothesize that added benefit of our parameter sharing scheme is to balance the capacity of generators and that of the discriminator/classifier.

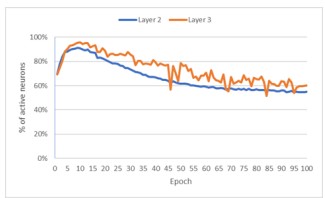
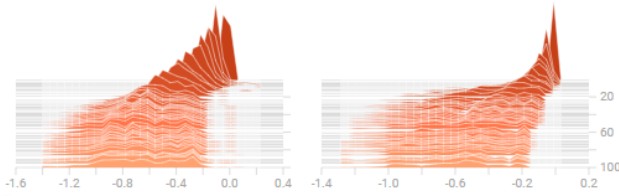

(a) % of active neurons in layer 2 and 3.

(b) Histogram of batch normalization centers in layer 2 (left) and 3 (right).

Figure 8: Observation of activate neuron rates and batch normalization centers in MGAN's generators trained on CIFAR-10.

**Experiment settings.** For the experiments on three large-scale natural scene datasets (CIFAR-10, STL-10, ImageNet), we closely followed the network architecture and training procedure of DCGAN. The specifications of our models trained on CIFAR-10, STL-10 48×48, STL-10 96×96 and ImageNet datasets are described in Tabs. (4, 5, 6, 7), respectively. "BN" is short for batch normalization and "BN center" is short for whether to learn batch normalization's center or set it at zero. "Shared" is short for parameter sharing among generators or between the classifier and the discriminator. Feature maps of 10/1 in the last layer for $C$ and $D$ means that two separate fully connected layers are applied to the penultimate layer, one for $C$ that outputs 10 logits and another for $D$ that outputs 1 logit. Finally, Figs. (9, 10, 11, 12, 13) respectively are the enlarged version of Figs. (3a, 3b, 3c, 4a, 4b) in the main manuscript.

Table 4: Network architecture and hyperparameters for the CIFAR-10 dataset.

| Operation | Kernel | Strides | Feature maps | BN? | BN center? | Nonlinearity | Shared? |
|---|---|---|---|---|---|---|---|
| $G(\mathbf{z}) : \mathbf{z} \sim$ Uniform $[-1, 1]$ | | | 100 | | | | |
| Fully connected | | | 4×4×512 | ✓ | × | ReLU | × |
| Transposed convolution | 5×5 | 2×2 | 256 | ✓ | × | ReLU | ✓ |
| Transposed convolution | 5×5 | 2×2 | 128 | ✓ | × | ReLU | ✓ |
| Transposed convolution | 5×5 | 2×2 | 3 | × | × | Tanh | ✓ |
| $C(\mathbf{x}), D(\mathbf{x})$ | | | 32×32×3 | | | | |
| Convolution | 5×5 | 2×2 | 128 | ✓ | ✓ | Leaky ReLU | ✓ |
| Convolution | 5×5 | 2×2 | 256 | ✓ | ✓ | Leaky ReLU | ✓ |
| Convolution | 5×5 | 2×2 | 512 | ✓ | ✓ | Leaky ReLU | ✓ |
| Fully connected | | | 10/1 | × | × | Softmax/Sigmoid | × |
| Number of generators | 10 | | | | | | |
| Batch size for real data | 64 | | | | | | |
| Batch size for each generator | 12 | | | | | | |
| Number of iterations | 250 | | | | | | |
| Leaky ReLU slope | 0.2 | | | | | | |
| Learning rate | 0.0002 | | | | | | |
| Regularization constants | $\beta = 0.01$ | | | | | | |
| Optimizer | Adam($\beta_1 = 0.5, \beta_2 = 0.999$) | | | | | | |
| Weight, bias initialization | $\mathcal{N}(\mu = 0, \sigma = 0.01), 0$ | | | | | | |

Table 5: Network architecture and hyperparameters for the STL-10 48×48 dataset.

| Operation | Kernel | Strides | Feature maps | BN? | BN center? | Nonlinearity | Shared? |
|---|---|---|---|---|---|---|---|
| $G\left(\mathbf{z}\right) : \mathbf{z} \sim \text{Uniform}\left[-1, 1\right]$ | | | 100 | | | | |
| Fully connected | | | 4×4×1024 | √ | × | ReLU | × |
| Transposed convolution | 5×5 | 2×2 | 512 | √ | × | ReLU | √ |
| Transposed convolution | 5×5 | 2×2 | 256 | √ | × | ReLU | √ |
| Transposed convolution | 5×5 | 2×2 | 128 | √ | × | ReLU | √ |
| Transposed convolution | 5×5 | 2×2 | 3 | × | × | Tanh | √ |
| $C\left(\mathbf{x}\right), D\left(\mathbf{x}\right)$ | | | 48×48×3 | | | | |
| Convolution | 5×5 | 2×2 | 128 | √ | √ | Leaky ReLU | √ |
| Convolution | 5×5 | 2×2 | 256 | √ | √ | Leaky ReLU | √ |
| Convolution | 5×5 | 2×2 | 512 | √ | √ | Leaky ReLU | √ |
| Convolution | 5×5 | 2×2 | 1024 | √ | √ | Leaky ReLU | √ |
| Fully connected | | | 10/1 | × | × | Softmax/Sigmoid | × |
| Number of generators | 10 | | | | | | |
| Batch size for real data | 64 | | | | | | |
| Batch size for each generator | 12 | | | | | | |
| Number of iterations | 250 | | | | | | |
| Leaky ReLU slope | 0.2 | | | | | | |
| Learning rate | 0.0002 | | | | | | |
| Regularization constants | $\beta = 1.0$ | | | | | | |
| Optimizer | Adam$(\beta_1 = 0.5, \beta_2 = 0.999)$ | | | | | | |
| Weight, bias initialization | $\mathcal{N}\left(\mu = 0, \sigma = 0.01\right), 0$ | | | | | | |

Table 6: Network architecture and hyperparameters for the STL96×96 dataset.

| Operation | Kernel | Strides | Feature maps | BN? | BN center? | Nonlinearity | Shared? |
|---|---|---|---|---|---|---|---|
| $G\left(\mathbf{z}\right) : \mathbf{z} \sim \text{Uniform}\left[-1, 1\right]$ | | | 100 | | | | |
| Fully connected | | | 4×4×2046 | √ | × | ReLU | × |
| Transposed convolution | 5×5 | 2×2 | 1024 | √ | × | ReLU | √ |
| Transposed convolution | 5×5 | 2×2 | 512 | √ | × | ReLU | √ |
| Transposed convolution | 5×5 | 2×2 | 256 | √ | × | ReLU | √ |
| Transposed convolution | 5×5 | 2×2 | 128 | √ | × | ReLU | √ |
| Transposed convolution | 5×5 | 2×2 | 3 | × | × | Tanh | √ |
| $C\left(\mathbf{x}\right), D\left(\mathbf{x}\right)$ | | | 32×32×3 | | | | |
| Convolution | 5×5 | 2×2 | 128 | √ | √ | Leaky ReLU | √ |
| Convolution | 5×5 | 2×2 | 256 | √ | √ | Leaky ReLU | √ |
| Convolution | 5×5 | 2×2 | 512 | √ | √ | Leaky ReLU | √ |
| Convolution | 5×5 | 2×2 | 1024 | √ | √ | Leaky ReLU | √ |
| Convolution | 5×5 | 2×2 | 2048 | √ | √ | Leaky ReLU | √ |
| Fully connected | | | 10/1 | × | × | Softmax/Sigmoid | × |
| Number of generators | 10 | | | | | | |
| Batch size for real data | 64 | | | | | | |
| Batch size for each generator | 12 | | | | | | |
| Number of iterations | 250 | | | | | | |
| Leaky ReLU slope | 0.2 | | | | | | |
| Learning rate | 0.0002 | | | | | | |
| Regularization constants | $\beta = 1.0$ | | | | | | |
| Optimizer | Adam$(\beta_1 = 0.5, \beta_2 = 0.999)$ | | | | | | |
| Weight, bias initialization | $\mathcal{N}\left(\mu = 0, \sigma = 0.01\right), 0$ | | | | | | |

Table 7: Network architecture and hyperparameters for the ImageNet dataset.

| Operation | Kernel | Strides | Feature maps | BN? | BN center? | Nonlinearity | Shared? |
|---|---|---|---|---|---|---|---|
| $G(\mathbf{z}) : \mathbf{z} \sim \text{Uniform}\,[-1, 1]$ | | | 100 | | | | |
| Fully connected | | | 4×4×512 | √ | × | ReLU | × |
| Transposed convolution | 5×5 | 2×2 | 256 | √ | × | ReLU | √ |
| Transposed convolution | 5×5 | 2×2 | 128 | √ | × | ReLU | √ |
| Transposed convolution | 5×5 | 2×2 | 3 | × | × | Tanh | √ |
| $C(\mathbf{x}), D(\mathbf{x})$ | | | 32×32×3 | | | | |
| Convolution | 5×5 | 2×2 | 128 | √ | √ | Leaky ReLU | √ |
| Convolution | 5×5 | 2×2 | 256 | √ | √ | Leaky ReLU | √ |
| Convolution | 5×5 | 2×2 | 512 | √ | √ | Leaky ReLU | √ |
| Fully connected | | | 10/1 | × | × | Softmax/Sigmoid | × |
| Number of generators | 10 | | | | | | |
| Batch size for real data | 64 | | | | | | |
| Batch size for each generator | 12 | | | | | | |
| Number of iterations | 50 | | | | | | |
| Leaky ReLU slope | 0.2 | | | | | | |
| Learning rate | 0.0002 | | | | | | |
| Regularization constants | $\beta = 0.1$ | | | | | | |
| Optimizer | $\text{Adam}(\beta_1 = 0.5, \beta_2 = 0.999)$ | | | | | | |
| Weight, bias initialization | $\mathcal{N}(\mu = 0, \sigma = 0.01), 0$ | | | | | | |

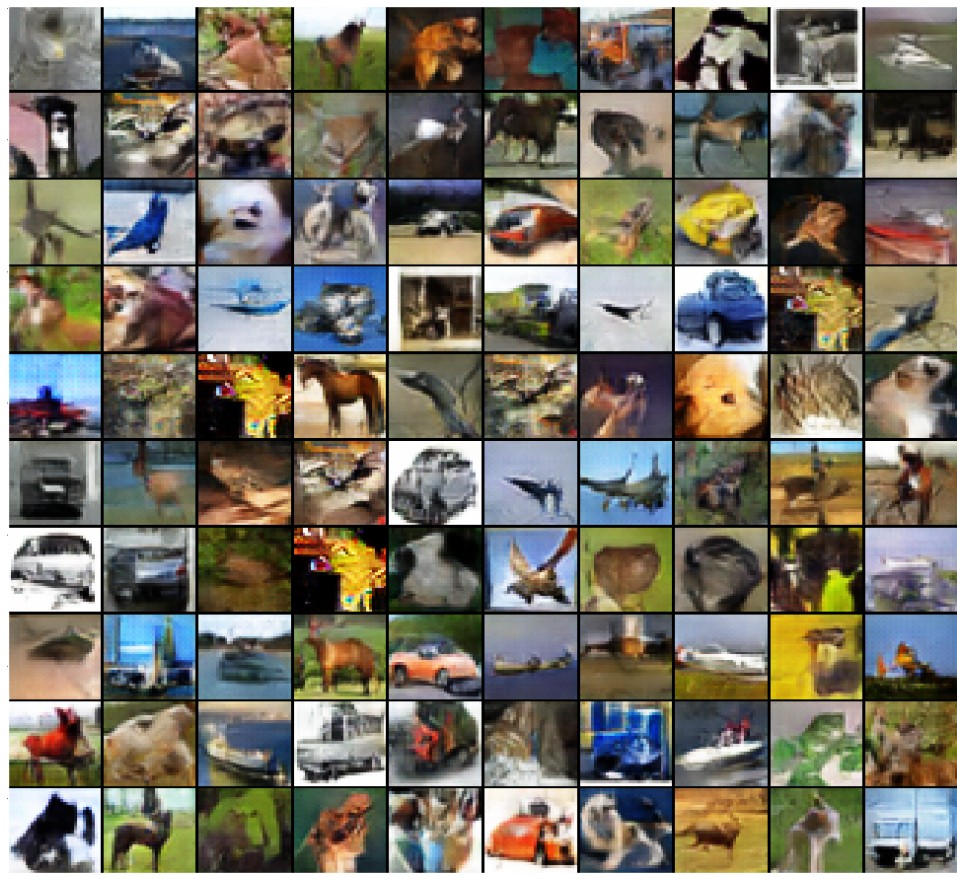

Figure 9: Images generated by MGAN trained on the CIFAR-10 dataset.

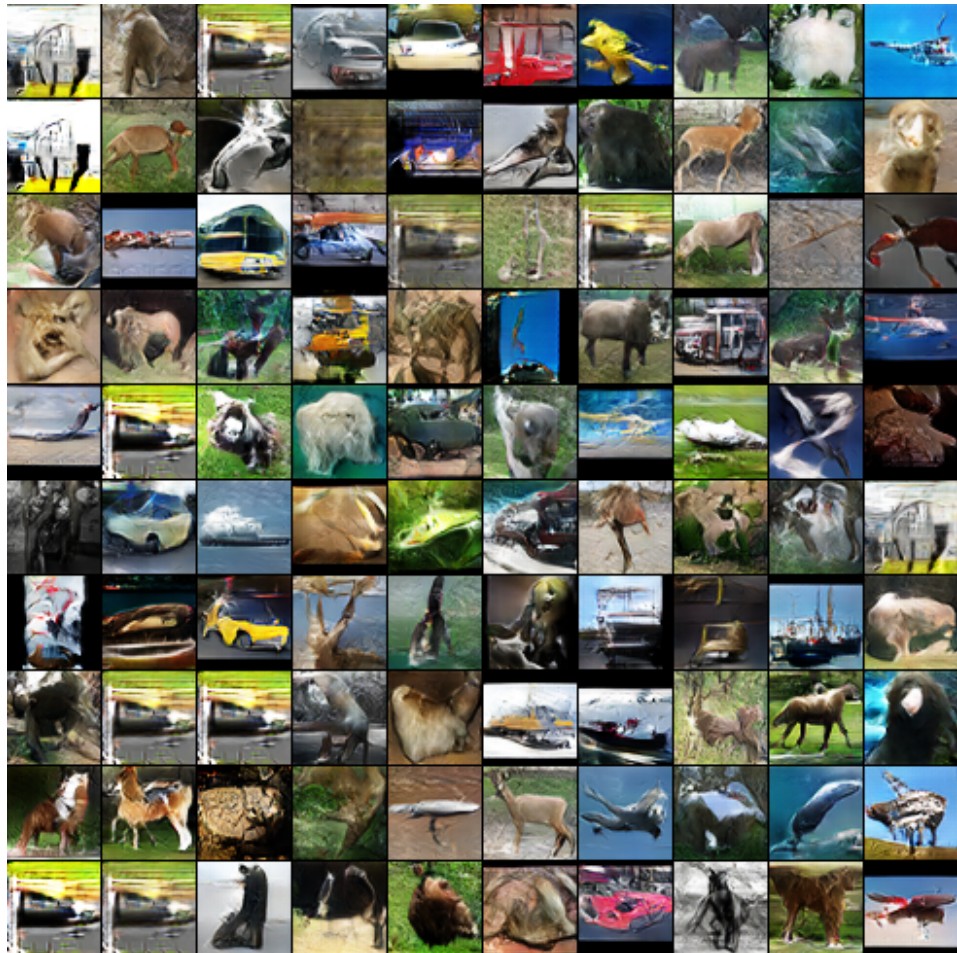

Figure 10: Images generated by MGAN trained on the rescaled 48×48 STL-10 dataset.

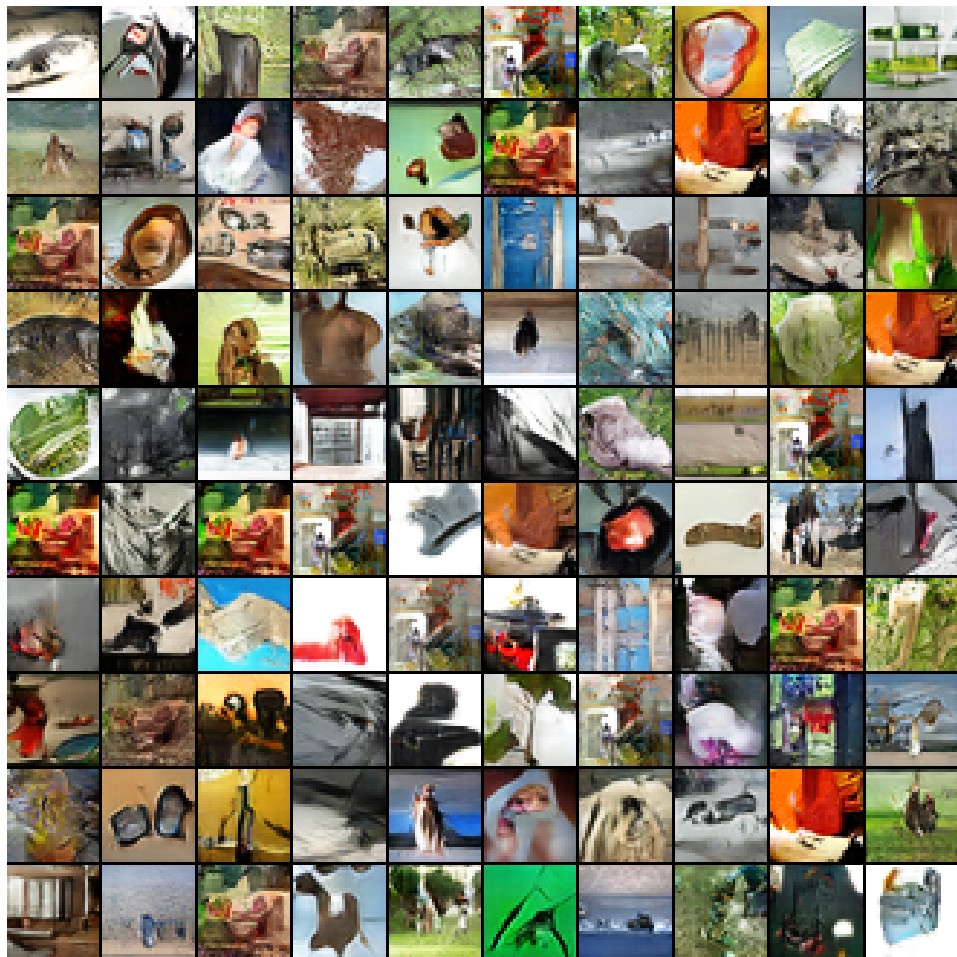

Figure 11: Images generated by MGAN trained on the rescaled 32×32 ImageNet dataset.

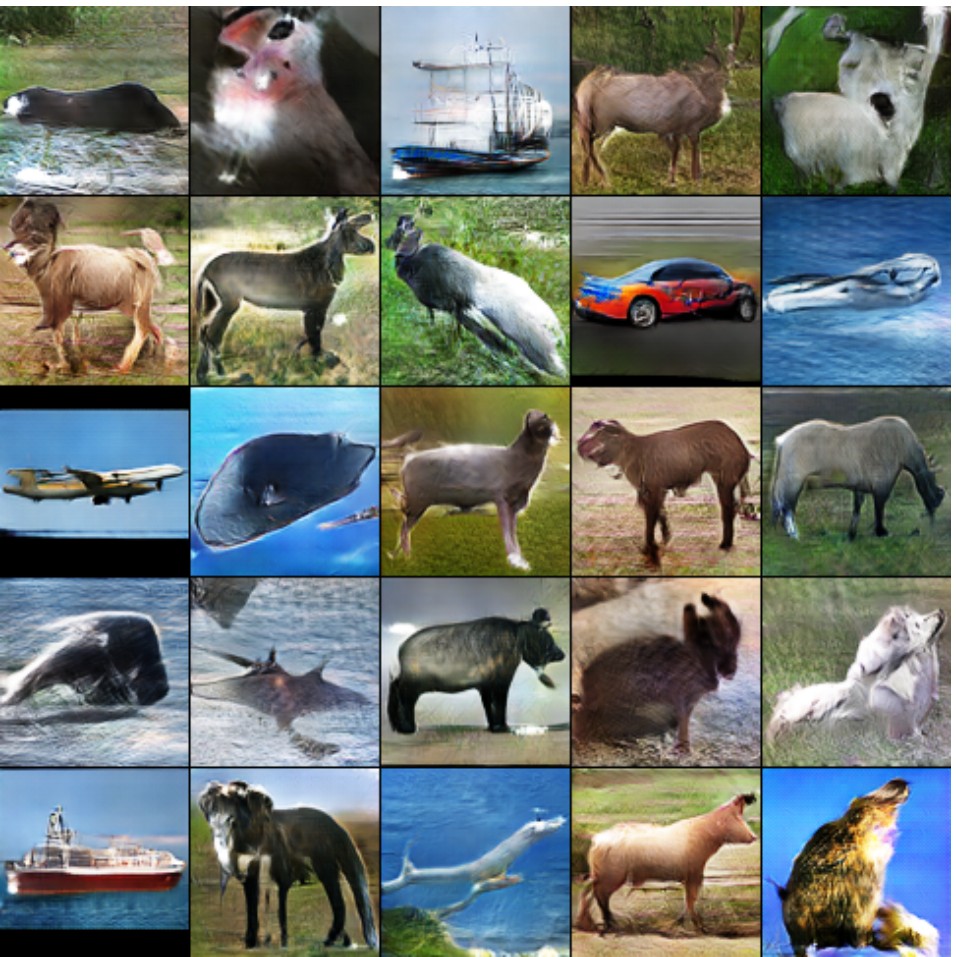

Figure 12: Cherry-picked samples generated by MGAN trained on the 96×96 STL-10 dataset.

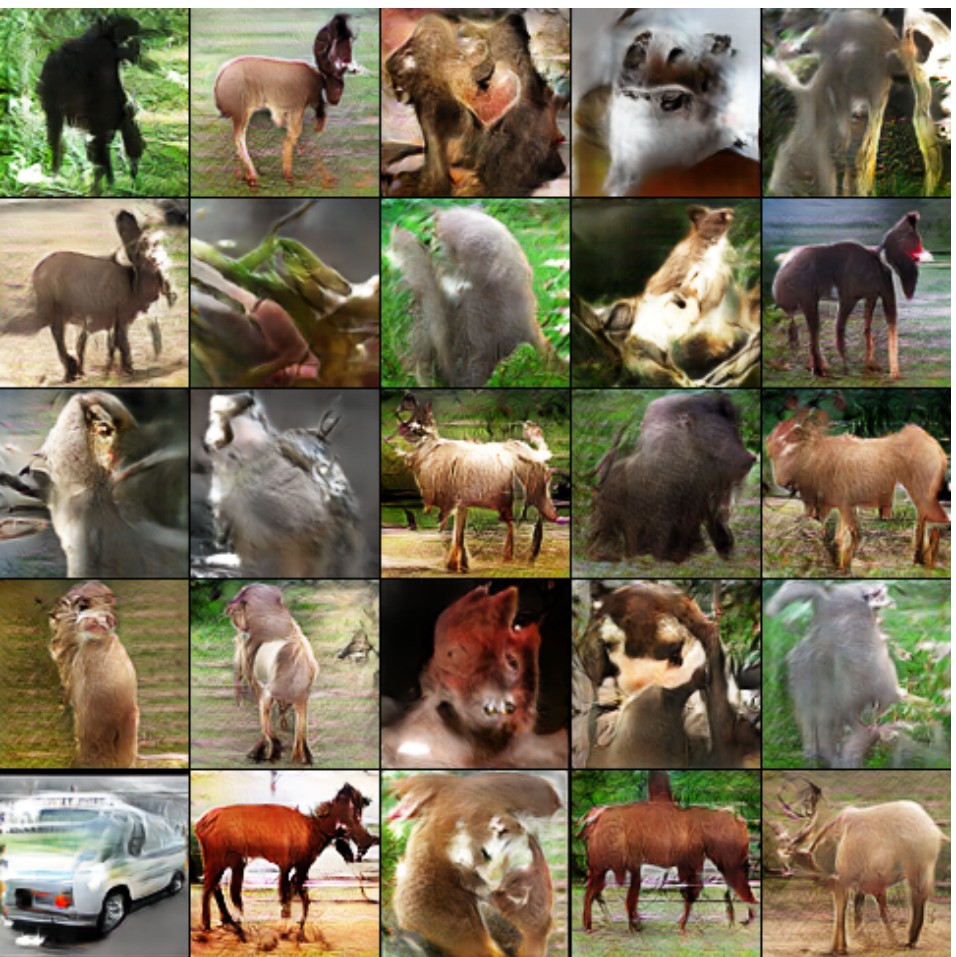

Figure 13: Incomplete, unrealistic samples generated by MGAN trained on the 96×96 STL-10 dataset.

