# OpenReview forum: "MGAN: Training Generative Adversarial Nets with Multiple Generators"
_ICLR.cc/2018/Conference — Accept (Poster)_

### Official Review · AnonReviewer2 · 2017-11-27
**Attempt at solving mode collapse just moves the problem.**

**Rating:** 5
**Confidence:** 4

**Review:**

The present manuscript attempts to address the problem of mode collapse in GANs using a constrained mixture distribution for the generator, and an auxiliary classifier which predicts the source mixture component, plus a loss term which encourages diversity amongst components.

All told the proposed method is quite incremental, as mixture GANs/multi-generators have been done before. The Inception scores are good but it's widely known now that Inception scores are a deeply flawed measure, and presenting it as the only quantitative measure in a manuscript which makes strong claims about mode collapse unfortunately will not suffice. If the generator were to generate one template per class for which the Inception network's p(y|x) had low entropy, the Inception score would be quite high even though the model had only memorized one image per class. For claims surrounding mode collapse in particular, evaluation against a parameter count matched baseline using the AIS log likelihood estimation procedure in Wu et al (2017) would be the gold standard. Frechet Inception distance has also been proposed which at least has some favourable properties relative to Inception score.

The mixing proportions are fixed to the uniform distribution, and therefore this method also makes the unrealistic assumption that modes are equiprobable and require an equal amount of modeling capacity. This seems quite dubious.

Finally, their own qualitative results indicate that they've simply moved the problem, with clear evidence of mode collapse in one of their mixture components in figure 5c, 4th row from the bottom. Indeed, this does nothing to address the problem of mode collapse in general, as there is nothing preventing individual mixture component GANs from collapsing.

Uncited prior work includes Generative Adversarial Parallelization of Im et al (2016). Also, if I'm not mistaken this is quite similar to an AC-GAN, where the classes are instead randomly assigned and the generator conditioning is done in a certain way; namely the first layer activations are the sum of K embeddings which are gated by the active mixture component. More discussion of this would be warranted.

Other notes:
- The introduction contains no discussion of the ill-posedness of the GAN game as it is played in practice.
- "As a result, the optimization order in 1 can be reversed" this does not accurately characterize the source of the issues, see, e.g. Goodfellow (2015) "On distinguishability criteria...".
- Section 3: the second last sentence of the third paragraph is vague and doesn't really say anything. Of course parameter sharing leverages common information. How does this help to train the model effectively?
- Section 3: Since JSD is defined between two distributions, it is not clear what JSD_pi(P_G1, P_G2, ...) refers to. The last line of the proof of theorem 2 leaps to calling this term a Jensen-Shannon divergence but it's not clear what the steps are; it looks like a regular KL divergence to me.
- Section 3: Also, is the classifier being trained to maximize this divergence or just the generator? I assume the latter.
- The proof of Theorem 3 makes unrealistic assumptions that we know the number of components a priori as well as their mixing proportions (pi).
- "... which further minimizes the objective value" -- it minimizes a term that you introduced which is constant with respect to your learnable parameters. This is not a selling point, and I'm not sure why you bothered mentioning it.
- There's no mention of the substitution of log (1 - D(x)) for -log(D(x)) and its effect on the interpretation as a Jensen-Shannon divergence (which I'm not sure was quite right in the first place)
- Section 4: does the DAE introduced in DFM really introduce that much of a computational burden?
- "Symmetric Kullback Liebler divergence" is not a well-known measure. The standard KL is asymmetric. Please define it.
- Figure 2 is illegible in grayscale.
- Improved-GAN score in Table 1 is misleading, as this was their no-label baseline. It's fine to include it but indicate it as such.

Update: many of my concerns were adequately addressed, however I still feel that calling this an avenue to "overcome mode collapse" is misleading. This seems aimed at improving coverage of the support of the data distribution; test log likelihood bounds via AIS (there are GAN baselines for MNIST in the Wu et al manuscript I mentioned) would have been more compelling quantitative evidence. I've raised my score to a 5.

---

> ### Author Response · Authors · 2017-12-16
> **Response to other Notes**
>
> **** Note 1: The introduction contains no discussion of the ill-posedness of the GAN game as it is played in practice.
>
> ==== Answer: We do not understand exactly what you meant by ill-posedness. Can please you further clarify this note?
>
> **** Note 2: "As a result, the optimization order in 1 can be reversed" this does not accurately characterize the source of the issues, see, e.g. Goodfellow (2015) "On distinguishability criteria...".
>
> ==== Answer: Here, we simply mentioned the issue discussed in The GAN tutorial (Goodfellow, 2016): “Simultaneous gradient descent does not clearly privilege min max over max min or vice versa. We use it in the hope that it will behave like min max but it often behaves like max min.”
>
> **** Note 3: Section 3: the second last sentence of the third paragraph is vague and doesn't really say anything. Of course parameter sharing leverages common informaNtion. How does this help to train the model effectively?
>
> ==== Answer: We discussed in Section 5.2, Model Architectures that our experiment showed that when the parameters are not tied between the classifier and discriminator, the model learns slowly and eventually yields lower performance.
>
> **** Note 4: Section 3: Since JSD is defined between two distributions, it is not clear what JSD_pi(P_G1, P_G2, ...) refers to. The last line of the proof of theorem 2 leaps to calling this term a Jensen-Shannon divergence but it's not clear what the steps are; it looks like a regular KL divergence to me.
>
> ==== Answer: The general definition of JSD is:
> JSD_pi(P_1, P_2, …P_n) = H(sum_{i=1..n} (pi_i * P_i)) -  sum_{i=1..n}(pi_i * H(P_i)
> Where H(P) is the Shannon entropy for distribution P. Due to limited space, we showed more details of the derivation of L(G_1:K) in Appendix B.
>
> **** Note 5: Section 3: Also, is the classifier being trained to maximize this divergence or just the generator? I assume the latter.
>
> ==== Answer: It is the latter. Based on Eq. 2, the classifier is trained to minimize its softmax loss, and based on the optimal solution for the classifier, the generators, by minimizing their objective function, will maximize the JSD divergence.
>
> **** Note 6: The proof of Theorem 3 makes unrealistic assumptions that we know the number of components a priori as well as their mixing proportions (pi). - "... which further minimizes the objective value" – it minimizes a term that you introduced which is constant with respect to your learnable parameters. This is not a selling point, and I'm not sure why you bothered mentioning it.
>
> ==== Answer: Please refer to our answer to comment 3.
>
> **** Note 7: There's no mention of the substitution of log (1 - D(x)) for -log(D(x)) and its effect on the interpretation as a Jensen-Shannon divergence (which I'm not sure was quite right in the first place)
>
> ==== Answer: We said in the end of Section 3: “In addition, we adopt the non-saturating heuristic proposed in (Goodfellow et al., 2014) to train G_{1:K} by maximizing log D(G_k (z)) instead of minimizing log D(1 - G_k (z)).”
>
> **** Note 8: Section 4: does the DAE introduced in DFM really introduce that much of a computational burden?
>
> ==== Answer: It was stated in Section 5.3, paragraph 2 in (Warde-Farley & Bengio, 2017) that: “we achieve a higher Inception score using denoising feature matching, using denoiser with 10 hidden layers of 2,048 rectified linear units each.” That means the DAE adds more than 40 million parameters.
>
> **** Note 9: “Symmetric Kullback Liebler divergence” is not a well-known measure. The standard KL is asymmetric. Please define it. - Figure 2 is illegible in grayscale.
>
> ==== Answer: Symmetric Kullback Liebler is the average of the KL and reverse KL divergence. As per your suggestion, we will define it in the paper. Regarding Figure 2, we tried different shapes for the real and generated data points, but due the small size if figure, they are just clusters of red and blue points. We will try different approaches to make the figure more legible.
>
> **** Note 10: Improved-GAN score in Table 1 is misleading, as this was their no-label baseline. It's fine to include it but indicate it as such.
>
> ==== Answer: We will take your advice and make it clear that Improve-GAN score in Table 1 is for the unsupervised version.

---

> ### Author Response · Authors · 2017-12-16
> **Response to Comment 4 and 5**
>
> **** Comment 4: Finally, their own qualitative results indicate that they've simply moved the problem, with clear evidence of mode collapse in one of their mixture components in figure 5c, 4th row from the bottom. Indeed, this does nothing to address the problem of mode collapse in general, as there is nothing preventing individual mixture component GANs from collapsing.
>
> ==== Answer: If we look carefully at samples shown in previously published papers (such as Figure 4 of the Improved GAN paper that showed samples generated by semi-supervised GAN trained on CIFAR-10 with feature matching), there are often broken samples that look similar.
>
> Solving mode collapse for a single-generator GAN is out of scope of this paper. As discussed in Introduction, we acknowledged the challenges of training a single generator, and therefore we took the multi-generator approach. We did not seek to improve within-generator diversity but instead improve among-generator diversity. The intuition is that GAN can be pretty good for narrow-domain datasets, so if a group of generators learns to partition the data space, and each of them focuses on a region of the data space, then they together can do a good job too. Finally, the use of a classifier to enforce divergence among generators makes our method relatively easy to integrate with other single-generator models that achieved improvement regarding the mode collapsing problem.
>
> **** Comment 5: Uncited prior work includes Generative Adversarial Parallelization of Im et al (2016). Also, if I'm not mistaken this is quite similar to an AC-GAN, where the classes are instead randomly assigned and the generator conditioning is done in a certain way; namely the first layer activations are the sum of K embeddings which are gated by the active mixture component. More discussion of this would be warranted.
>
> ==== Answer: Generative Adversarial Parallelization (GAP) trains many pairs of GAN, periodically swap the discriminators (generators) randomly, and finally selects the best GAN based on GAM evaluation. When we discussed methods in the multi-generator approach, we focused on mixture GAN and as a result neglected GAP. It is fair to discuss GAP as an approach to reduce the mode collapsing problem.
>
> In AC-GAN, the label information and the noise are concatenated and then fed into the generator network. In our model, generators have different weights in the first layer, so they are mapped to the first hidden layer differently. MGAN and AC-GAN both add the log-likelihood of the correct class to the objective function, but the motivation is very different. Our idea started by asking how to force generators to generate different data, while AC-GAN's motivation is to leverage the label information from training data. So, the two works are totally independent and happens to share some similarities. Our paper focuses on unsupervised GAN, so we did not discuss semi-supervised methods.

---

> ### Author Response · Authors · 2017-12-16
> **Response to Comment 3 and Note 6**
>
> **** Comment 3: The mixing proportions are fixed to the uniform distribution, and therefore this method also makes the unrealistic assumption that modes are equiprobable and require an equal amount of modeling capacity. This seems quite dubious.
>
> **** Note 6: The proof of Theorem 3 makes unrealistic assumptions that we know the number of components a priori as well as their mixing proportions (pi). - "... which further minimizes the objective value" – it minimizes a term that you introduced which is constant with respect to your learnable parameters. This is not a selling point, and I'm not sure why you bothered mentioning it.
>
> ==== Answer: Our theorem 3 shows that by means of maximizing the divergence among the generated distributions p_G_k ( ⋅ ) , in an ideal case, our proposed MGAN can recover the true data distribution wherein each p_G_k describes a mixture component in this data distribution. Although this theorem gives more insightful understanding of our MGAN as well as its behavior, it requires a strict setting wherein we need to specify the number of mixtures and the mixing proportions a priori. Stating this theorem, we want to emphasize that maximizing the divergence among the generated distributions p_G_k is an efficient way to encourage the generators to produce diverse data that can occupy multiple modes in the real data. Moreover, since GAN requires training a single generator that can cover multiple data modes, it is much harder to train, and always ends up with missing of data modes. In contrast, our MGAN aims at training each generator to cover one or a few data modes, hence being easier to train, and reducing the missed data modes. In addition, due to the fact that each generator can cover some data modes, the number of generators K can be less than the number of data modes as shown in Figure 6 wherein samples generated from 3 or 4 generators can well cover a mixture of 8 Gaussians.
>
> Given the fact that we are learning from the empirical data distribution, we develop a further theorem to clarify that if we wish to learn the mixing proportion π, the optimal solution is the uniform distribution. The idea is that the optimal generators will learn to partition the empirical data into K disjoint sets of roughly equal size, and each generator approximates a set. In addition, due to the fact that the discrete distribution p_A_k is well-approximated by a continuous generator G_k, the data points in each A_k occupies several groups or clusters. Again, Figure 6 illustrates this point. In Figure 6b, each of the 2 generators (yellow and blue) covers 4 modes. In Figure 6c, one generator (dark green) covers 2 modes and the other two generators (yellow and blue) covers 3 modes. In Figure 6d, each of the four generators (yellow, blue, dark green and dodger blue) cover 2 modes.
>
> For details of our theorem, please refer to this link: https://app.box.com/s/jjr5kt69uxbr0aikrm0d9cdp2jj95wa0

---

> ### Author Response · Authors · 2017-12-16
> **Response to Comment 2**
>
> **** Comment 2: The Inception scores are good but it's widely known now that Inception scores are a deeply flawed measure, and presenting it as the only quantitative measure in a manuscript which makes strong claims about mode collapse unfortunately will not suffice. If the generator were to generate one template per class for which the Inception network's p(y|x) had low entropy, the Inception score would be quite high even though the model had only memorized one image per class. For claims surrounding mode collapse in particular, evaluation against a parameter count matched baseline using the AIS log likelihood estimation procedure in Wu et al (2017) would be the gold standard. Frechet Inception distance has also been proposed which at least has some favourable properties relative to Inception score.
>
> ==== Answer: We chose Inception Score because at the time we set up our experiment, it was the most widely accepted metrics, so it would be easier for us to compare with many baselines. We did acknowledge that any quantitative metric has its weakness and Inception Score is no exception. Therefore, we included a lot of samples in the paper and looked at them from different angle. It can be noticed that our samples, in terms of quality, are far better than those shown in previously published papers. In addition, we looked at samples generated by each of the generators to check whether they trap Inception Score by memorizing a few examples from each class. We saw no sign of trapping as samples generated by each generator were diverse, especially on diverse datasets such as STL-10 or ImageNet. Therefore, we believe that our method achieved higher Inception Score than single-GAN methods not because it trapped the score, but because each of the generators learned to model a different subset of the training data. As a result, our generated samples are more diverse and at the same time more visually appealing. For the mentioned reasons, we strongly believe the use of Inception Score in our experiment to evaluate our proposed method is valid and plausible.
>
> As per your suggestion, we looked for GAN baselines using the AIS loglikelihood, but we found no GAN baseline. Regarding Frechet Inception (FID) distance, our model got an FID of 26.7 for Cifar-10. Some baselines we collected from (Heusel et al., 2017) are 37.7 for the original DCGAN, 36.9 for DCGAN using Two Time-scale Update rule (DCGAN + TTUR), 29.3 for WGAN-GP (Gulrajani, 2017) FID of 29.3, and 24.8 for WGAN-GP using TTUR. It is noteworthy that lower FID is better, and that the base model for MGAN is DCGAN. Therefore, in terms of FID, MGAN (26.7) is 28% better than DCGAN (37.7) and DCGAN using TTUR (36.9) and is 9% better than WGAN-GP (29.3), which uses ResNet architecture. This example further shows evidence that our proposed method helps to address the mode collapsing problem.

---

> ### Author Response · Authors · 2017-12-16
> **Response to comment 1**
>
> We gratefully thank the reviewer for the detailed and valuable comments and notes. It took us a while to thoughtfully answer all the comments, and the following are our answers. Due to the limited number of characters per comment, we will answer in several posts:
>
> **** Comment 1: All told the proposed method is quite incremental, as mixture GANs/multi-generators have been done before.
>
> ==== Answer: As discussed in related work, there are previous attempts following the multi-generators approach, but they are different from our proposed method. Mix+GAN is totally different as it's based on the min-max theorem and set up mixed strategies for both generators and discriminators. AdaGAN train generators sequentially in a manner similar to AdaBoost, thus having some disadvantages as we discussed. MAD-GAN, at a first glance, looks somewhat similar to our proposed method in terms of model design, but there are some key differences. First, it uses a multi-class discriminator, which outputs D_k(x) as the probability that x generated by G_k for k = 1, 2, … K, and D_{K+1}(x) as the probability that x came from the training data. The gradient signal for each generator k comes from the loss function E_{x~p_G_k}[log (1 - D_{k+1}(x)], which is similar to that in a standard GAN. So, it might be vulnerable to the issue discussed in the Improved GAN paper: “Because the discriminator processes each example independently, there is no coordination between its gradients, and thus no mechanism to tell the outputs of the generator to become more dissimilar to each other.” Our proposed method is distinguished in the use of a classifier to enforce JSD divergence among generators. In addition, the use of a separate classifier makes our method easier to integrate with other single-generator GAN models. There is also extension to our method that do not apply to MAD-GAN. We can use the classifier to cluster the train data, and then further train each generator in a different cluster.
>
> In terms of performance, our method is far superior than Mix+GAN both in terms of Inception Scores and sample quality. The AdaGAN only presents experiment on MNIST. MAD-GAN mostly performed experiment on narrow-domain datasets, and they did not report any quantitative data on diverse datasets and did not release code as well.

---

### Official Review · AnonReviewer3 · 2017-11-27
**review for MGAN**

**Rating:** 7
**Confidence:** 5

**Review:**

Summary:

The paper proposes a mixture of  generators to train GANs. The generators used have tied weights except the first layer that maps the random codes is generator specific, hence no extra computational cost is added.


Quality/clarity:

The paper is well written and easy to follow.

clarity: The appendix states how the weight tying is done , not the main paper, which might confuse the reader, would be better to state this weight tying that keeps the first layer free in the main text.

Originality:

 Using multiple generators for GAN training has been proposed in many previous work that are cited in the paper, the difference in this paper is in weight tying between generators of the mixture, the first layer is kept free for each generator.

General review:

- when only the first layer is free between generators, I think it is not suitable to talk about multiple generators, but rather it is just a multimodal prior on the z, in this case z is a mixture of Gaussians with learned covariances (the weights of the first layer). This angle should be stressed in the paper, it is in fine, *one generator* with a multimodal learned prior on z!

- Taking the multimodal z further , can you try adding a mean to be learned, together with the covariances also? see if this also helps?

- in the tied weight case, in the synthetic example, can you show what each "generator" of the mixture learn? are they really learning modes of the data?

- the theory is for general untied generators, can you comment on the tied case? I don't think the theory is any more valid, for this case, because again your implementation is one generator with a multimodal z prior.  would be good to have some experiments and  see how much we loose for example in term of inception scores, between tied and untied weights of generators.

---

> ### Author Response · Authors · 2017-12-16
> **Response**
>
> We gratefully thank the reviewer for the thoughtful and insightful comments. It took us a while to answer all the reviews as well as to run additional experiments as suggested. Our answers are the following:
>
> **** Comment 1: when only the first layer is free between generators, I think it is not suitable to talk about multiple generators, but rather it is just a multimodal prior on the z, in this case z is a mixture of Gaussians with learned covariances (the weights of the first layer). This angle should be stressed in the paper, it is in fine, *one generator* with a multimodal learned prior on z!
>
> ==== Answer: The first hidden layer actually has 4x4x512 = 8,192 dimensions (for Cifar-10). So, untying weights in the first layer effectively maps the noise prior to a different distribution in R^8192 (with a different mean and covariances) for each generator. So, our proposed method is different from a GAN with a multimodal prior.
>
> **** Comment 2: taking the multimodal z further , can you try adding a mean to be learned, together with the covariances also? see if this also helps?
>
> ==== Answer: We tried to learn the mean and covariance of the prior for each generator, but the result was not much different from the standard GAN.
>
> **** Comment 3: in the tied weight case, in the synthetic example, can you show what each "generator" of the mixture learn? are they really learning modes of the data?
>
> ==== Answer: Following your suggestion, we revised figure 6 so that data points generated by different generators have different colors. As you can see, generators learned different modes of the data.
>
> **** Comment 4: the theory is for general untied generators, can you comment on the tied case? I don't think the theory is any more valid, for this case, because again your implementation is one generator with a multimodal z prior. would be good to have some experiments and see how much we loose for example in term of inception scores, between tied and untied weights of generators.
>
> ==== Answer: In theory, tying weights will add constraints to the optimization of the objective function for G_{1:K} in Eq. 4. For example, if we tie weights in all layers and generators differ only in the mean and variance of the noise prior, the result was similar to the standard GAN like we reported in comment 2. Untying weights in the first layer, however, achieved good results like we discussed in the paper. Finally, as per your request, we conducted experiments without parameter sharing. Surprisingly, when we trained 4 generators without parameter sharing and each generator has 128 feature maps in the penultimate layer, the model failed to learn. The model even failed to learn when we set beta to 0. When we reduced the number of feature maps in the penultimate layer for each generator to 32, they managed to learn and achieved an Inception Score of 7.42. So, we hypothesize that added benefit of our parameter sharing scheme is to balance the capacity of generators and that of the discriminator/classifier.

---

> > ### Comment · AnonReviewer3 · 2018-01-11
> > **relation to multimodal priors/ untying the weights**
> >
> > Thanks for your reply.
> >
> > Comment 1: if we take z standard gaussian. Let (W_j,b_j) be the first Layer that is untied. we have W_jz remains gaussian with mean b_j and covariance W_jW_j^{\top}. We can think of W_jz as a multimodal prior that feed to the shared generator. so the implementation given in the paper is indeed a multimodal prior (degenerate multivariate gaussians in R^{8192}). It is true that this not standard multimodal prior in low dimension, but since the gaussians in R^{8192} are degenerate, they are still supported on a low dimensional subspace.
> >
> > Comment 4: The experiment on untying the weights and the effect of regularization of having smaller bottleneck is interesting, and maybe worth adding to the paper.

---

> > > ### Author Response · Authors · 2018-01-22
> > > **More on multimodal priors/ untying the weights**
> > >
> > > We gratefully thank the reviewer for the insightful response!
> > >
> > > Comment 1: If we view this as graphical model where hidden units and the output G(z) are random variables, then the implementation in the paper can be seen as a multimodal prior. However, hidden units and the output G(z) are learned deterministic functions of z, so each G_k(z) implies a different distribution. Therefore, it is still appropriate to see the implementation as a mixture.
> > >
> > > Comment 4: We will follow your suggestion and add the experiment result to the paper.

---

### Official Review · AnonReviewer1 · 2017-11-29

**Rating:** 6
**Confidence:** 3

**Review:**

MGAN aims to overcome model collapsing problem by mixture generators. Compare to traditional GAN, there is a classifier added to minimax formulation. In training, MGAN is optimized towards minimizing the Jensen-Shannon Divergence between mixture distributions from generator and data distribution. The author also present that using MGAN to achive state-of-art results.

The paper is easy to follow.

Comment:

1. Seems there still no principle to choose correct number of generators but try different setting. Although most parameters of generators are shared, the result various.
2. Parameter sharing seems is a trick in MGAN model. Could you provide experiment results w/o parameter sharing.

---

> ### Author Response · Authors · 2017-12-16
> **Response**
>
> We gratefully thank reviewers for the insightful comments. We have endeavored to address as much as we can, including running additional experiments as suggested, thus it has taken us a while.
>
> **** Comment 1: Seems there still no principle to choose correct number of generators but try different setting. Although most parameters of generators are shared, the result various.
>
> ==== Answer: We agree that we don’t have any principle to choose the correct number of generators for our proposed model, as choosing the correct number of clusters for Gaussian mixture model (GMM) and other clustering methods. If we wish to specify an appropriate number of generators automatically, we would need to go for a Bayesian nonparametric extension, similarly to going from GMM to Dirichlet Process Mixtures. Within the scope of this work, our motivation is that GAN works pretty well on narrow-domain dataset but poorly on diverse dataset; So, if we can efficiently train many generators while enforcing divergence among them, they can work well too. In general, more generators tend to work better.
>
> **** Comment 2: Parameter sharing seems is a trick in MGAN model. Could you provide experiment results w/o parameter sharing.
>
> ==== Answer: We did experiment without parameters sharing among generators and found an interesting behavior. When we trained 4 generators without parameter sharing and each generator has 128 feature maps in the penultimate layer, the model failed to learn. The model even failed to learn when we set beta to 0. When we reduced the number of feature maps in the penultimate layer for each generator to 32, they managed to learn and achieved an Inception Score of 7.42. So, we hypothesize that added benefit of parameter sharing is to help balance the capacity of generators and that of the discriminator/classifier.

---

### Author Response · Authors · 2017-12-16
**Revision**

A revision has been posted with some minor changes. We added the definition of symmetric Kullback-Leibler in Section 5.1, clarified in Table 1's caption that all models in the table are trained in a unsupervised manner, and changed the Figure 6 so that data generated by each generator have a different color.

---

### Public Comment · (anonymous) · 2018-02-16
**MGAN does not solve mode collapse**

The classifier implicitly reduce the region each GAN can explore, making the mode collapse problem more serious. This phenomenon is similar to what happened in conditional GAN. If we consider the mode counting method introduced in "Do GANs learn the distribution? Some Theory and Empirics" by Arora et al. then when mode collapse occurs with high probability then the number of modes is only in order of the batch size. From figure 3, we can see that the probability of mode collapse for batch size of 100 is 100% which implies that there are only hundreds of modes in the model distribution. From figure 5, we can see that a generator is collapsed completely to only 2 modes. The high inception scores might be the (unwanted) result of mode collapse.

The paper is also filled with unnecessary proofs of convergence for their new loss function while proof for more general loss functions already exist. I don't see the point of proving that MGAN converge to a equilibrium. The authors better provide an analysis of their loss function: Why and how much it helps to alliviate mode collapse. However, acording to the analysis in the previous paragraph, MGAN makes mode collapse problem more serious.

The fairness in evaluation is another concern.  The authors included very weak baselines such as MIX + GAN with DCGAN architecture and claimed that their method out performed simple MIX + GAN protocol.

---

> ### Author Response · Authors · 2018-02-21
> **response**
>
> Thank you for your comment!
>
> As for the fairness in evaluation, we included not only MIX + GAN but also the latest baselines at the time we wrote the paper. At the time, most models employed the DCGAN architecture like our MGAN did.
>
> Our idea is that each generator focuses on a different region of the data space so the mixture of them will improve mode coverage. Regarding your claim that high inception scores might be a result of mode collapse, we calculated the Frechet Inception Distance. MGAN attained a FID of 26.7, which is better than most recent baselines. To our knowledge, among models that use architectures similar to DCGAN, only SN-GAN has a better FID (25.5). Finally, we recently trained MGAN with the ResNet architecture and got a FID of 21.9, which is also close to SN-GAN ResNet verion's 21.7. These quantitative results show that MGAN reduces the mode collapse problem.

---

> > ### Public Comment · (anonymous) · 2018-02-22
> > **Lower FID scores does not means less mode collapsing**
> >
> > As stated in "GANs Trained by a Two Time-Scale Update Rule Converge to a Local Nash Equilibrium" Heusel et al., FID "captures the similarity of generated images to real ones better than the Inception Score". The authors did not make any claim that lower FID means less mode collapsing.
> >
> > Mode collapse can be seen clearly in figure 3 of your paper. The following images are the same:
> > Row 4, col 9; row 5, col 3; row 7, col 4 of figure 3a
> > Row 1, col 1; row 2, col 1 and row 10, col 1; row 10, col 2 of figure 3b
> > Row 6, col 1; row 6, col 3 and row 2, col 8; row 4, col 10; row 9, col 9 of figure 3c.
> >
> > Mode collapse are present in all of your generated samples with batch size 100. That implies that there are only hundreds of modes in your model distribution. Normal DCGAN can capture thoundsands to hundreds thoundsand of modes. Therefore, the combined capacity of genrators in MGAN is lower than a single GAN trained with normal loss. It show that your model has more serious mode collapse. As mentioned in my previous comment, this is the result of implicit label introduced by the classifier. The same phenomenon has been observed for conditional GAN.

---

> > > ### Author Response · Authors · 2018-02-25
> > > **response**
> > >
> > > The is no perfect measure to evaluate mode collapse. Still, FID is a good measure as it captures the similarity of generated images to real ones. Good FID does confirm the merit of our method.
> > >
> > > The examples you mentioned are actually images generated by one generator that was collapsed as we mentioned in figure 5. When we removed collapsed generator from the mixture, we saw no duplicates and the score improved. In some recent experiments, we considered some ways to avoid this problem. We tried adding new generators gradually instead of training all 10 generators from the beginning and we didn't see this problem. We also did not observe this problem with the ResNet architecture.

---

> > > > ### Public Comment · (anonymous) · 2018-02-26
> > > > **reponse**
> > > >
> > > > Thank you for your reponse. I understand your approach now.

---

> > ### Public Comment · (anonymous) · 2018-02-22
> > **Comparison should be carried on networks of the same size and/or architectures**
> >
> > MIX + GAN used networks of very small size just to demonstrate that mixtures lead to more stable training. You should include the result of MIX + GAN with the same number of parameters, not the result reported in their paper.

---

> > > ### Author Response · Authors · 2018-02-25
> > > **MIX+GAN uses 5 times more parameters than DCGAN**
> > >
> > > The MIX+GAN paper says: "Our method is applied to DCGAN and WassersteinGAN Arjovsky et al. [2017], and throughout, mixtures of 5 generators and 5 discriminators are used. At rst sight the comparison DCGAN v.s. MIX+DCGAN seems unfair because the latter uses 5 times the capacity of the former, with corresponding penalty in running time per epoch. To address this, we also compare our method with larger versions of DCGAN with roughly the same number of parameters, and we found the former is consistently better than the later, as detailed below."
> > >
> > > The reason we compare only to the MIX+WGAN because it's the only model trained without label in the paper: "Table 1: Inception Scores on CIFAR-10. Mixture of DCGANs achieves higher score than any single-component DCGAN does. All models except for WassersteinGAN variants are trained with labels."

---

### Public Comment · (anonymous) · 2018-02-27
**When one generator is collapsed, does that mean a class disappear?**

If each generator models a different region then when a generator is collapsed does it mean that the corresponding region is missing in the model distribution? In your CIFAR10 example, some classes seem missing in the generated samples.

What is your opinion about this very simple approach to forcing each generator to model a different region of the target distribution:
1. Divide the training data into $n$ subsets
2. Train a generator on each subset

---

> ### Author Response · Authors · 2018-02-27
> **response**
>
> When a generator is collapsed, it is likely due to the divergence force. It is also possibly due to the limitation of the training method. When we recently tried using ResNet or adding new generator gradually, no generator collapsed.
>
> If we randomly divide the dataset into K subsets, it's likely that the diversity of the dataset is retained in each subset, while the size of each subset is K times less than the size of the full dataset. Therefore, each generator won't learn as well as it would when trained on the full dataset. The idea of our method is that generators will learn to divide the data space themselves, and each generator specializes in a different region.

---

> > ### Public Comment · (anonymous) · 2018-02-27
> > **Wasserstein loss for discriminator**
> >
> > Thank you for your detailed response. The original GAN loss is the cause of mode collapse in many scenarios. Have you done any experiments with Wasserstein loss?

---

> > > ### Author Response · Authors · 2018-02-27
> > > **response**
> > >
> > > Thank you for the question. This is a great question!
> > >
> > > We did conduct a crude experiment by employing the WGAN-GP model and add a classifier, which is a softmax classifier utilizing the last layer of the critic. However, the result wasn't better than MGAN using ResNet and JSD loss. We observed that the softmax loss got stuck around 2.0 (for 10 classes, entropy loss for random guesses is about ln(10) = 2.3). So, it seems that the critic performs a different task than a discriminator so parameter sharing is not effective. Using an independent classifier without parameter sharing might work but that's not efficient.
> > >
> > > An interesting idea for experiment might be using WS-distance instead of JSD to force divergence among generators. For example, the critic can return K additional outputs, each of which helps estimate the WS-distance between a generator and the rest. WS-distance is weaker than f-divergence, so it does not vanish like f-divergence and might be more effective. We haven't conducted the experiment yet because enforcing Lipschitz for this idea using WGAN-GP isn't elegant. Employing Spectral Normalization for this idea might be interesting because Spectral Normalization is an elegant way (but perhaps too restrictive?) to enforce Lipschitz. But we are curious about how well SN-GAN works for WS-distance. For some reasons, SN-GAN's authors used a hinge loss instead of critic loss although Spectral Normalization would be perfect for WGAN.

---

> > > > ### Public Comment · (anonymous) · 2018-02-28
> > > > **Does mode collapse happen when Wasserstein distance is used?**
> > > >
> > > > When the network is trained with Wasserstein loss, do you observe mode collapse at any of the generators?

---

> > > > > ### Author Response · Authors · 2018-03-13
> > > > > **response**
> > > > >
> > > > > Sorry that we did not notice your follow-up. No, we did not observe mode collapse at any of the generators. However, the generators did not diverge well because the softmax classifier using the last layer of the critic was not effective.

---

> > > > > > ### Public Comment · (anonymous) · 2018-03-16
> > > > > > **Does that mean the total capacity of M generators is upper bounded**
> > > > > >
> > > > > > If we increase the diversity of generators, then each generator capture a more compact region of the target distribution. If we expand the region that each generator can capture, we loose the diversity of generators. Does it means that the total capacity of M generator is upper bounded. As suggested by Arora et al., the number of modes actually depends on the capacity of the discriminator, not generators. We need strong discriminator to produce strong generators. Adding a classifier does not increase the capacity of the discriminator. I am aware that the classifier force each generator to model a different region of the target distribution. However, it does not help generators to cover more modes. An xperiment that could verify the effectiveness of your algorithm is:
> > > > > > - Building a large discriminator D and train two GANs: D and a single G_0; and D and mixture of G_i whose total of number of parameters is the same as G_0. If the mixture covers more mode then your approach is truly effective. Otherwise, MGAN is just a way of dividing the dataset.

---

> > > > > > > ### Author Response · Authors · 2018-03-30
> > > > > > > **response**
> > > > > > >
> > > > > > > A very good paper on this topic is: https://openreview.net/forum?id=Hk9Xc_lR-
> > > > > > > According to the paper, high capacity of the discriminator helps discrimination but hurt generalization.
> > > > > > > We actually did the experiment you suggested in our experiment. Remember that we used parameter sharing, which doesn't add many parameters compared to DCGAN with the same number of feature maps. We tried with 192 feature maps in the penultimate layer of the discriminator. We also tried with generators with 32 feature maps in the penultimate layer. The results weren't much different from the one we reported, which is significantly better than DCGAN.

---

### Decision · Program_Chairs · 2018-01-29
**ICLR 2018 Conference Acceptance Decision**

**Decision:**

Accept (Poster)

**Comment:**

This paper presents an analysis of using multiple generators in a GAN setup, to address the mode-collapse problem. R1 was generally positive about the paper, raising the concern on how to choose the number of generators, and also whether parameter sharing was essential. The authors reported back on parameter sharing, showing its benefits yet did not have any principled method of selecting the number of generators. R2 was less positive about the paper, pointing out that mixture GANs and multiple generators have been tried before. They also raised concern with the (flawed) Inception score as the basis for comparison. R2 also pointed out that fixing the mixing proportions to uniform was an unrealistic assumption. The authors responded to these claims, clarifying the differences between this paper and the previous mixture GAN/multiple generator papers, and reporting FID scores. R3 was generally positive, also citing some novelty concerns similar to that of R2. I acknowledge the authors detailed responses to the reviews (in particular in response to R2) and I believe that the majority of concerns expressed have now been addressed. I also encourage the authors to include the FID scores in the final version of the paper.